# Assessing the performance of methods for cell clustering from single-cell DNA sequencing data

**Rituparna Khan, Xian Mallory**[ORCID]*

Department of Computer Science, Florida State University, Tallahassee, Florida, United States of America

* xfan2@fsu.edu

## Abstract

### Background

Many cancer genomes have been known to contain more than one subclone inside one tumor, the phenomenon of which is called intra-tumor heterogeneity (ITH). Characterizing ITH is essential in designing treatment plans, prognosis as well as the study of cancer progression. Single-cell DNA sequencing (scDNAseq) has been proven effective in deciphering ITH. Cells corresponding to each subclone are supposed to carry a unique set of mutations such as single nucleotide variations (SNV). While there have been many studies on the cancer evolutionary tree reconstruction, not many have been proposed that simply characterize the subclonality without tree reconstruction. While tree reconstruction is important in the study of cancer evolutionary history, typically they are computationally expensive in terms of running time and memory consumption due to the huge search space of the tree structure. On the other hand, subclonality characterization of single cells can be converted into a cell clustering problem, the dimension of which is much smaller, and the turnaround time is much shorter. Despite the existence of a few state-of-the-art cell clustering computational tools for scDNAseq, there lacks a comprehensive and objective comparison under different settings.

### Results

In this paper, we evaluated six state-of-the-art cell clustering tools–SCG, BnpC, SCClone, RobustClone, SCITE and SBMClone–on simulated data sets given a variety of parameter settings and a real data set. We designed a simulator specifically for cell clustering, and compared these methods' performances in terms of their clustering accuracy, specificity and sensitivity and running time. For SBMClone, we specifically designed an ultra-low coverage large data set to evaluate its performance in the face of an extremely high missing rate.

### Conclusion

From the benchmark study, we conclude that BnpC and SCG's clustering accuracy are the highest and comparable to each other. However, BnpC is more advantageous in terms of

**Data Availability Statement:** The simulator described in this study, as well as the Python scripts that generated the simulated data sets, the instructions to run the six methods along with the post-processing scripts are publicly available at

https://github.com/compbiofan/
clusteringBenchmark. The tools being tested were
downloaded from the GitHub pages: SCG (https://
github.com/Roth-Lab/scg), SCClone (https://
github.com/qasimyu/scclone), BnpC (https://
github.com/cbg-ethz/BnpC), RobustClone (https://
github.com/ucasdp/RobustClone), SCITE (https://
github.com/cbg-ethz/SCITE) and SBMClone
(https://github.com/raphael-group/SBMClone). The
real data CRC2 used for our experiments was
downloaded from https://www.ncbi.nlm.nih.gov/
sra under accession number SRP074289.

**Funding:** R.K. and X.M. were supported by the
startup funding from Florida State University. The
funders had no role in study design, data collection
and analysis, decision to publish, or preparation of
the manuscript.

**Competing interests:** The authors have declared
that no competing interests exist.

running time when cell number is high (> 1500). It also has a higher clustering accuracy than
SCG when cluster number is high (> 16). SCClone's accuracy in estimating the number of
clusters is the highest. RobustClone and SCITE's clustering accuracy are the lowest for all
experiments. SCITE tends to over-estimate the cluster number and has a low specificity,
whereas RobustClone tends to under-estimate the cluster number and has a much lower
sensitivity than other methods. SBMClone produced reasonably good clustering (V-mea-
sure > 0.9) when coverage is > = 0.03 and thus is highly recommended for ultra-low cover-
age large scDNAseq data sets.

## Author summary

Cancer cells evolve by gaining new mutations. Different cancer cells may gain different
mutations. Thus even inside the tumor of one cancer patient, there could be multiple
clones of cancer cells, each having its unique set of mutations. Characterization of the
clonality of a tumor can improve cancer treatment and prognosis. Single-cell DNA
sequencing, or scDNAseq, refers to the DNA sequencing technology that can sequence
each cell separately. ScDNAseq has been widely used to characterize the clonality of a
tumor. In this study, we benchmarked six existing computational tools that characterize
the clonality of the cancer cells, which are SCG, SCClone, BnpC, RobustClone, SCITE and
SBMClone. To accomplish this study, we modified our simulator and simulated eight sets
of simulation data for testing the first five methods, and one set of simulation data for test-
ing SBMClone considering that SBMClone was specifically designed for highly sparse
data set. We also tested SCG, SCClone, BnpC and RobubstClone on a real data set CRC2.
We recommend SCG and BnpC due to their high accuracy and low running time. In the
face of highly sparse data, we highly recommend SBMClone, the only method that can
deal with such type of data.

## Introduction

Cancer progresses with acquired mutations [1–3]. During this process, different cancer cells
inside the same tumor may acquire different sets of mutations, leading to a notoriously diffi-
cult-to-solve problem, the intra-tumor heterogeneity (ITH) [4–14]. In particular, the co-exis-
tence of multiple subclones of cancer cells leads to the malignancy of cancer, and even
metastasis, which is responsible for 90% of the death of cancer patients [15]. ITH also is a con-
founding factor in treatment and prognosis due to the lack of a comprehensive knowledge of
the clonality in the tumor. Therefore, it is important to fully characterize the clonality in a can-
cer genome to guide the prescription of drug or chemotherapy, to obtain an accurate progno-
sis, as well as to gain a better understanding of the mechanism of cancer progression.

Single-cell DNA sequencing (scDNAseq) data, due to its sequencing each cell separately, is
unique and advantageous in understanding the clonality and ITH [16, 17]. Technical chal-
lenges of scDNAseq lie in the whole genome amplification (WGA) of the tiny amount of the
DNA (6pg) in the cell by 3 to 9 orders of magnitude [18] for sequencing library construction
[19]. Particularly, multiple displacement amplification (MDA)-based approach [20–22] gener-
ates scDNAseq data that have high depth of the sequencing, facilitating the detection of single
nucleotide variants (SNVs) [19, 23, 24]. However, MDA-based WGA suffers from high allele
dropout (ADO) rate. ADO is a technical failure to provide measurement of both alleles, and is

the leading cause of false negatives (FNs) in SNV detection [19, 25]. On the other hand, the infidelity of polymerase enzymes may lead to false positive (FPs) SNVs in SNV detection [23]. Such false positive SNVs are extremely excessive in C:G>T:A transitions [26]. Finally, MDA-based WGA technologies render low amplification breadth over the genome [26], leading to severe missing calls in SNV detection. Technically, FNs are generated due to the lack of variant-supporting reads whereas missing calls are due to the lack of reads, both variant- and reference-supporting, covering the mutation site.

In the past decade, a myriad of bioinformatics tools have been developed to tackle the ITH problem specifically designed for scDNAseq data while considering its unique error profile. These bioinformatics tools mainly have three functions, inferring the evolutionary history of cancer cells, characterizing the clonal composition of the cells, and inferring the genotype of all the mutation sites for each cell. Some of the tools infer all of the three, whereas some only aim at the latter two. In general, if a tool infers the evolutionary history of cancer cells, it can also characterize the clonality of the cells and infer the genotype of the mutation sites. Those tools that can jointly infer the evolutionary history and characterize the clonality of cancer cells include but are not limited to SCITE [27], OncoNEM [28], Sifit [29], SiCloneFit [30], SASC [31], SPhyR [32], and GRMT [33].

The output of these tools typically consists of an evolutionary tree delineating the cancer evolution history. The class of the tree varies among phylogenetic trees in which each node represents a single cell, mutation trees in which each node represents a new mutation, and clonal lineage trees in which each node represents a subclone of single cells [34]. No matter which class of the tree is used, the single cells can always be placed on the tree. Some tools, like SCITE [27], added an additional step to attach them to the leaf nodes.

A post-processing step is then required to convert the placement of the cells on the tree to the clustering of the cells. In particular, on a clonal lineage tree or a mutation tree, the cells attached to the same node are assigned to the same cluster. On a phylogenetic tree in which each leaf node represents a sequenced cell, clustering of the cells has to be done by cutting the branches at a certain level on the tree so that the cells attached to the same branch can be assigned to the same cluster.

To obtain the consensus genotype of the mutations for each cluster, one can traverse the path on the tree from the root to the corresponding leaf node (or the node below which a branch is cut) that the cells of the same cluster are attached to. This consensus genotype profile also serves as the corrected genotype profile for all the cells assigned to the same cluster.

While it is desirable to have both the cancer evolutionary history and the clonal composition inside a tumor, it is computationally expensive to infer the cancer evolutionary tree due to the vast number of trees that the algorithms have to search and evaluate. The ever-increasing number of cells in single-cell sequencing field makes it even more challenging to infer an evolutionary tree in terms of the cost of computational resources as well as the turn-around time [25]. Clinically, what a medical provider needs is mainly the number of clones in the tumor for the prognosis purpose, not necessarily the whole evolutionary tree, although the cancer evolutionary tree is helpful in understanding how cancer cells evolve. In such sense, clustering the cells without inferring the evolutionary tree becomes essential due to its less demanding computational resources and fast turn-around time [25]. As a matter of fact, in the case when the evolutionary history of the cancer development is of interest, separating the clustering of the cells and the inference of the cancer evolutionary history is becoming a future trend, considering the increasing number of single cells sequenced in each sample [25, 35]. In this way, not only can the genotype errors be corrected during the cell clustering stage, but also the lineage tree is searched based on the subclones (clusters of the cells) instead of the individual cells.

In all, a separate cell clustering step provides a tremendously smaller search space for the search of lineage tree.

The problem of clustering, the procedure of identifying similar groups given a set of data, has been addressed by many machine learning tools. Some of them require the knowledge of the number of clusters (in our context, it is the number of clones of cells), for example, K-means [36], mixture model using EM algorithm [37], hierarchical clustering [38], etc. Others do not require the knowledge of the number of the clusters, for example, mean-shift [39], Density-Based Spatial Clustering of Applications with Noise (DBSCAN) [40], and more advanced ones such as shared nearest neighbor (SNN) [41] algorithm. In recent years clustering has also been related with the problem of community detection where each data point (in our case, cell) represents a node in a graph, and the similarity between every pair of data points represents an edge between two nodes. Community detection algorithms, such as Infomap [42] and Louvain [43] method, can be re-purposed for the detection of clusters given the distance or similarity between each pair of the cells in terms of their mutation profiles.

Despite the plethora of clustering methods in the realm of machine learning, only a few clustering methods are available that are specifically designed for scDNAseq data. Yet the conventional clustering method cannot be directly applied to scDNAseq data because of scDNAseq's unique error profile. Specifically, the false negative, false positive and missing rates shall be incorporated in the design of the clustering algorithm. The few scDNAseq-based tools that aim at only the cell clustering, not the lineage tree inference, are SCG [44], BnpC [25], SCClone [45], SBMClone [46], RobustClone [47] and ARCANE-ROG [48]. Here ARCANE-ROG is omitted in our benchmark due to that it is not open-source. Of SCG, SCClone, BnpC, SBMClone and RobustClone, SCG [44] is the first tool published to cluster scDNAseq data. It uses mean-field variational inference to infer the probabilities of cell assignment and the corrected genotype. SCG is unique in that it was designed for multiple data types, including binary genotypes, where the mutation is either existent or absent; ternary genotypes, where the mutation could be homozygous variant, heterozygous variant and homozygous reference; as well as the data with doublets in which two cells are captured and sequenced together as if they were one cell. When SCG is run in the doublet mode, it is called D-SCG. BnpC [25] is a fully non-parametric Bayesian method that can handle noisy data in a large quantity. BnpC is scalable to handle a large number of clusters as well as thousands of cells. SCClone [45] uses a probability mixture model to cluster single cells into distinct clusters and uses an Expectation Maximization (EM) algorithm to infer the model parameters. Since mixture model requires the prior knowledge of the number of clusters, SCClone performs the EM algorithm for each possible cluster number and searches for the optimal cluster number. SCClone's results depend heavily on the initial values of the model parameters such as false positive and false negative rates. SBMClone [46] is designed to infer clusters of cells for ultra-low coverage scDNAseq data. The application of SBMClone includes those data whose coverage is as low as 0.03x. Due to the low coverage, most SNV sites' data is missing whereas the missing rate can reach approximately 99.89%. SBMClone uses stochastic block model (SBM) that clusters both cells and mutations. The output of SBMClone is a block matrix where each row indicates a cluster of cells and each column indicates a cluster of mutations. RobustClone [47] uses extended robust principal component analysis (RPCA) to recover the genotypes of cells. It then clusters cells using Louvain–Jaccard method and reconstructs the subclonal evolutionary trees. It is a model-free method that can be applied to both single-cell data intended for SNV detection and CNA detection. It has been claimed to be a fast tool that is scalable for large data sets. It is worth noting that celluloid [49] and AMC [50] are two mutation clustering methods for scDNAseq data. Since this study focuses on clustering cells instead of clustering mutations, we do not include them in the benchmark and discussion.

Despite the importance of cell clustering for scDNAseq data, there has not been a comprehensive benchmark on the existing cell clustering methods. Therefore their performance is unclear under different settings. Moreover, despite the fact that SCClone benchmarked four above-mentioned methods, SCG, SCClone, BnpC, and RobustClone on their simulated data, we found that the parameters of the methods could have been better tuned to improve the performance of the tools, with the help of the correspondences with the authors. Thus, the conclusion SCClone made regarding the comparison of these methods is not necessarily reliable. On the other hand, although SCClone's simulation covered a few important parameters, the simulation is still not comprehensive. What is especially interesting but missing is the effect of the contrast of the cluster sizes on the clustering results. This is relevant because small subclones are more challenging to characterize due to the lack of representative cells. Moreover, it is important to compare these four methods with tree-based methods such as SCITE [27], as tree-based methods can render the clustering of the cells as well. Finally, given the enormous amount of ultra-low coverage scDNAseq data that were originally sequenced for the CNA detection, the evaluation of SBMClone [46] that was specifically designed for such data is essential for the sake of re-purposing the data for new discoveries.

We therefore designed simulation studies which allowed us to comprehensively benchmark the existing scDNAseq cell clustering methods. A comprehensive list of clustering-based methods have been included for comparison, which are SCG, SCClone, BnpC, RobustClone and SBMClone. In addition, we also included the state-of-the-art tree-based method SCITE [27] for comparison with the five aforementioned clustering-based methods. Thus, in total there are six methods involved in our benchmark study, which are SCG, SCClone, BnpC, RobustClone, SBMClone and SCITE. We evaluated these methods in terms of their accuracy of clustering, accuracy of the inference of the genotypes, and the consumption of computational resources. In addition, given a variety of simulated data sets, we also evaluated their accuracy of estimating the number of clones, an important metric of the performance of the methods. Our simulator has various parameters to mimic the realistic data, which allowed us to investigate the advantages and disadvantages of the six methods under different settings. In particular, what is new but essential in this benchmark study is the simulation of varying variance of cluster sizes. We modified the tree construction algorithm, the Beta splitting model used in SimSCSnTree [51, 52], so that we can tune a parameter in the simulator to increase or decrease the variance of the cluster sizes. The benchmark of the resulting simulated data set is indicative of the robustness of the clustering method for the cases when the contrast of the cluster sizes is big or small. In the following sections, we describe our simulator in detail, followed by the design of the simulation experiments, i.e., the list of data sets generated by varying parameters in the simulator. We briefly describe the comparison metrics that will be applied to measure the results of all six methods, as well as the parameter setting of these six methods. We then discuss the simulation results for each data set, followed by the discussion of the results of a real data set CRC2 [53]. We summarize the study with the discussion of observations from the benchmark study, linking the algorithm design and the clustering performance for each method, and by making recommendation of which tool to use under which setting in our conclusion.

## Results

### Benchmark of simulation experiments

**Simulation results for the data intended for SNV detection.** Five methods, SCG, SCClone, BnpC, RobustClone, and SCITE, were benchmarked on the data intended for SNV detection. We did not apply SBMClone to these data sets because SBMClone was specifically

designed for large but ultra-low coverage data set. The following showed the performance of each of the aforementioned five methods for the eight varying variables in the simulation.

First, we varied the false positive rates in the simulated data while keeping all other parameters at their default values. As expected, all five methods' performance dropped when false positive rate increased (Fig 1). Among the five methods, BnpC was the most stable method whose performance only slightly dropped with the increase of false positive rates. SCG's performance was next to BnpC, with slightly lower sensitivity compared with BnpC ($\sim$99.4% compared with $\sim$99.8%). SCClone was the next, whose sensitivity dropped from 99.9% to 99.1%, and V-measure dropped from 0.99 to 0.87 when the false positive rate increased from 0.001 to 0.05. RobustClone's specificity was comparable to BnpC, SCG and SCClone, but it failed to maintain as high the sensitivity and V-measure. Its sensitivity was >5% lower than BnpC, SCG and SCClone, and its V-measure was around $\sim$0.9, to be compared with $\sim$0.99 for BnpC and SCG. SCITE generated results with reasonable sensitivity (ranging between 94.3% and 96.5%) and specificity (ranging between 97.4% and 99.2%). However, SCITE's V-measure was much worse than BnpC, SCG, SCClone and RobustClone, dropping from 0.74 to 0.55 with the increase of false positive rate.

Of all five methods, RobustClones' running time was the least for all false positive rates (<10s for all false positive rates), whereas all other methods required at least hundreds of seconds to finish the task. SCG's speed was the next, ranging from $\sim$450s to $\sim$780s. SCClone had a significant increase of the running time with the increase of false positive rate, from

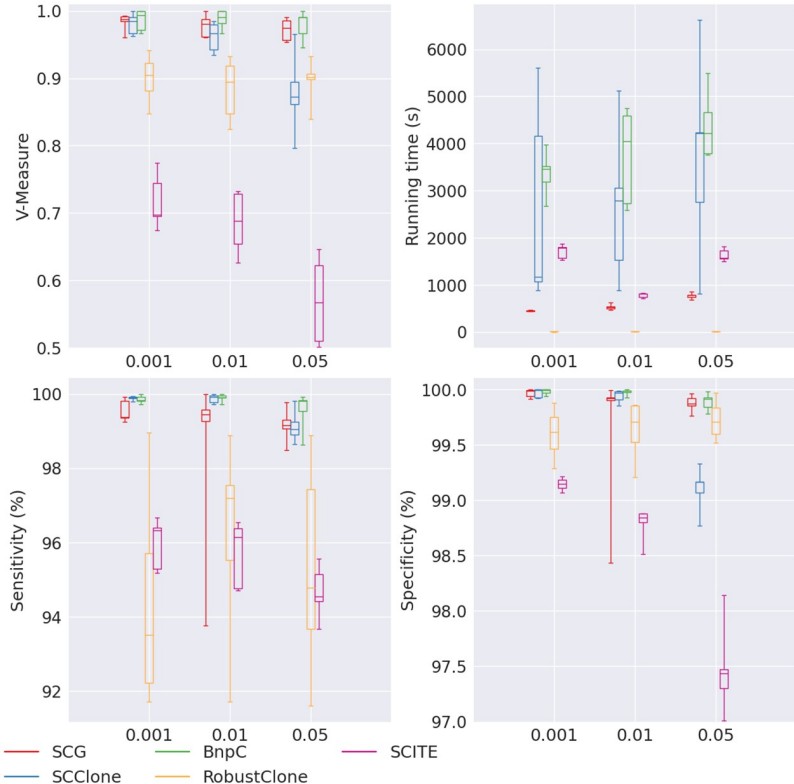

**Fig 1. Performance of SCG (red), SCClone (blue), BnpC (green), RobustClone (yellow), and SCITE (purple) for varying false positive rate, the values of which are shown on the x-axes.** The upper left, upper right, bottom left and bottom right panels are the V-measure, running time in seconds, genotyping sensitivity and genotyping specificity, respectively.

1170s to 4,213s. BnpC required more running time in this experiment than SCG, ranging from 3,461s to 4,215s. SCITE's running time ranged from 795s to 1,522s, and surprisingly, SCITE's running time decreased with the increase of false positive rate. Such trend, however, did not demonstrate SCITE's advantage considering that SCITE's V-measure decreased tremendously with the increase of false positive rate.

We further varied the false negative rate from 0.1 to 0.4. Almost all methods had decreasing V-measure, sensitivity and specificity when the false negative rate increased (Fig 2). Of the five methods, SCG, SCClone and BnpC had the highest and comparable V-measure (all > 0.96), sensitivity (all > 99%) and specificity (all > 99.8%). RobustClone had a tremendous drop of V-measure when false negative rate increased, from 0.94 to 0.45. While RobustClone's specificity remained high for all false negative rates (> 99.7%), its sensitivity decreased from 97.3% to 59% with the increase of false negative rate. The observation that RobustClone had a high specificity but much lower sensitivity in this experiment is consistent with what we observed in the experiment of varying false positive rate. SCITE's V-measure was much lower than SCG, SCClone and BnpC, dropping from 0.74 to 0.58 with the increase of false negative rate. Like RobustClone, SCITE's specificity remained high for all false negative rates (> 98.2%), but its sensitivity dropped with the increase of false negative rate (from 98.1% to 78.9%).

In terms of the running time, like what we observed in the experiment of varying false positive rate, RobustClone was still the fastest of all five methods, taking < 10s to finish the tasks for all false negative rates. SCG, SCClone and BnpC's running time increased with the increase

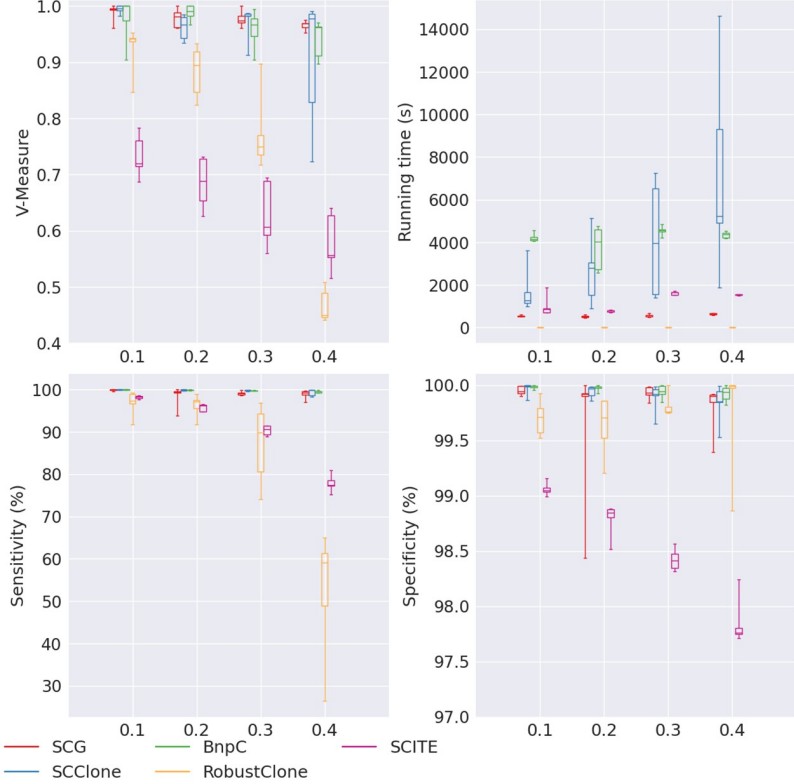

**Fig 2. Performance of SCG (red), SCClone (blue), BnpC (green), RobustClone (yellow), and SCITE (purple) for varying false negative rates, the values of which are shown on the x-axes.** The upper left, upper right, bottom left and bottom right panels are the V-measure, running time in seconds, genotyping sensitivity and genotyping specificity, respectively.

of false negative rate. SCG's running time was ∼600s and BnpC's was ∼4,000s. SCClone's running time increased from 1,281s to 5,246s, showing that SCClone's running time was very sensitive to the varying false negative rate. SCITE's running time did not show any trend and ranged between 812s and 1,768s.

When missing rate varied from 0.2 to 0.3, we noticed a slight drop of performance for SCG and BnpC in terms of their V-measure, sensitivity and specificity (Fig 3). We did not observe a drop of V-measure, sensitivity and specificity for SCClone when missing rate increased. RobustClone's specificity was robust to the varying missing rate, and was comparable to SCG, SCClone and BnpC (all > 99.7%), but its sensitivity decreased from 97.1% to 94.7%, and V-measure decreased from 0.89 to 0.84 when the missing rate increased from 0.2 to 0.3. SCITE also had worse performance for larger missing rate. SCITE's V-measure was the worst among all five methods, ∼0.67 for the varying missing rate.

SCG's running time slightly increased, from 522s to 584s, when the missing rate increased. BnpC was much slower than SCG, ranging between 4,046s and 4,177s. It took RobustClone < 10s to finish the task, thus RobustClone stayed the fastest of the five methods. Interestingly, SCClone showed a reverse trend of the running time when the missing rate increased, i.e., its median running time decreased from 2787s to 1122s when the missing rate increased from 0.2 to 0.3. SCITE's running time increased from 812s to 1,689s with the increasing missing rate.

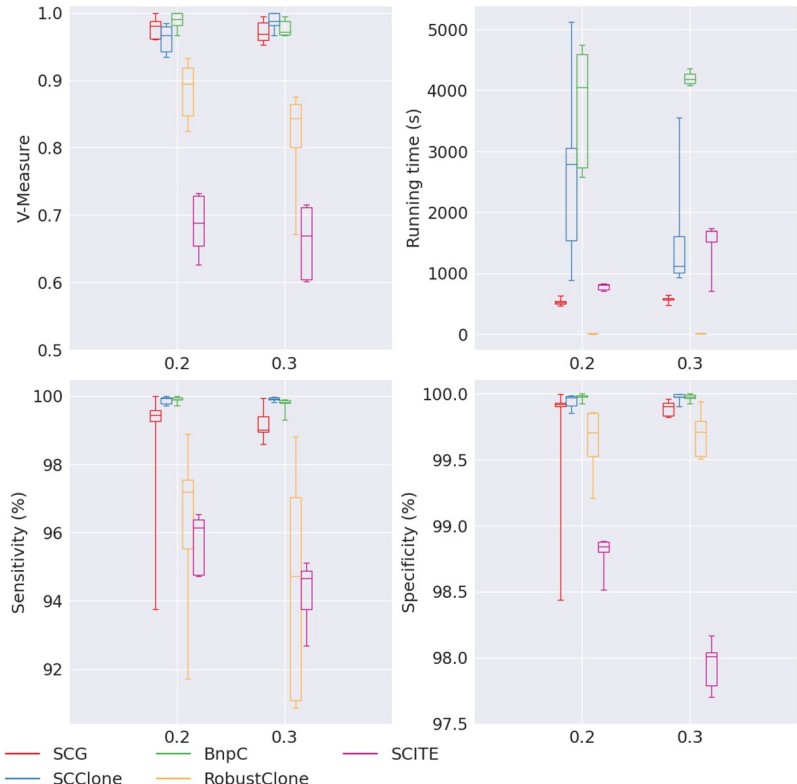

**Fig 3. Performance of SCG (red), SCClone (blue), BnpC (green), RobustClone (yellow), and SCITE (purple) for varying number of missing rates, the values of which are shown on the x-axes.** The upper left, upper right, bottom left and bottom right panels are the V-measure, running time in seconds, genotyping sensitivity and genotyping specificity, respectively.

Next, we varied the number of cells from 100 to 1500. SCG, SCClone, BnpC and SCITE showed increasing V-measure, sensitivity and specificity when the number of cells increased (Fig 4). SCG, SCClone and BnpC maintained a high sensitivity (all > 98.1%), high specificity (all at 100%), and high V-measure (all > 0.97). SCITE, however, had much lower sensitivity (ranging between 89.4% and 97.8%), lower specificity (ranging between 95.3% and 99.6%) and much lower V-measure (ranging between 0.65 and 0.7). RobustClone did not show any trend with the increasing cell number. RobustClone's specificity remained high (> 99%) and was comparable to SCG, SCClone and BnpC's. However, RobustClone's sensitivity was lower, ranging between 90.3% and 97.1%. Its V-measure was also lower compared with SCG, SCClone and BnpC, ranging from 0.82 to 0.93.

Except RobustClone, all methods' running time increased when the number of cells increased. Of all, BnpC's running time growth rate was the lowest: it only doubles (from 2,793s to 5,712s) when the number of cells increased from 100 to 1500, showing that BnpC is scalable to large data set. SCG's running time increased from 65s to 7,178s (by 110 times) and SCClone's running time increased from 108s to 12,418s (by 114 times) when the number of cells increased from 100 to 1500. Thus these two algorithms' running time growth rate was more than linear of the cell number. SCITE's running time increased from 392s to 5,132s (by 13 times), showing that SCITE's running time growth rate is about linear of the cell number. RobustClone's running time continued to remain as the lowest, all < 10s regardless of the cell number.

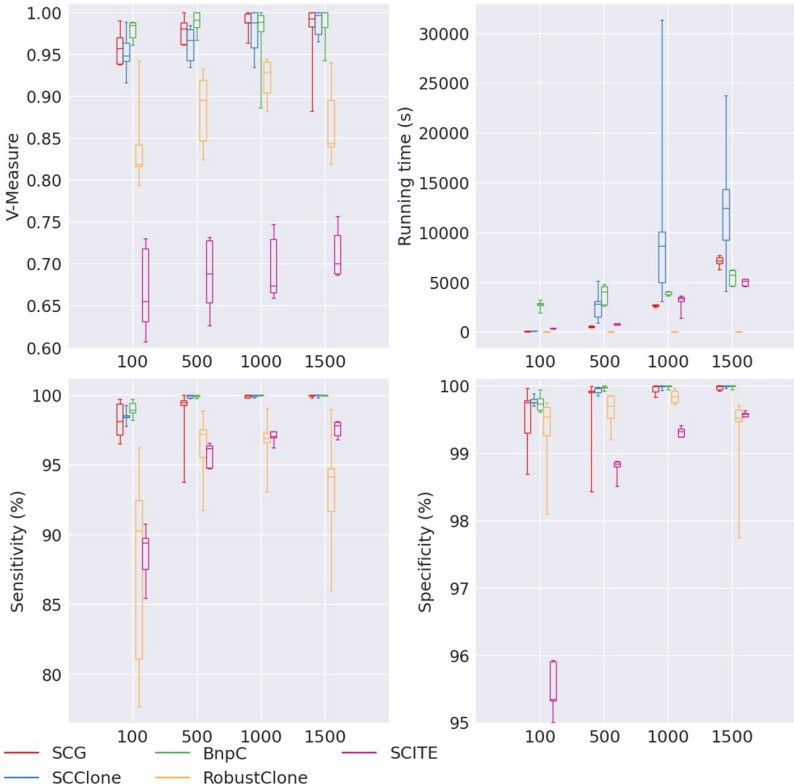

**Fig 4. Performance of SCG (red), SCClone (blue), BnpC (green), RobustClone (yellow), and SCITE (purple) for varying number of cells, the values of which are shown on the x-axes.** The upper left, upper right, bottom left and bottom right panels are the V-measure, running time in seconds, genotyping sensitivity and genotyping specificity, respectively.

For the experiment of varying number of mutation, we noticed an increasing trend for SCG, SCClone, BnpC and RobustClone's V-measure when the number of mutations increased, whereas SCITE showed a reverse trend (Fig 5). SCG, SCClone and BnpC's V-measure stayed the highest of the five methods for all numbers of mutations. RobustClone was extremely sensitive to the number of mutations, and it performed the worst (V-measure <0.6) when mutation number was small. We also noticed that when the mutation number was as small as 50, RobustClone's sensitivity was the lowest (~80%). SCITE's specificity decreased with the increase of the mutation number, and its sensitivity slightly increased with the increase of the mutation number. This is probably due to that SCITE over-corrected false negative mutations when the mutation number was high.

All methods' running time increased when the number of mutations increased. Robust-Clone's running time was still the least (~10s) for all numbers of mutations. Of all methods, SCITE's running time growth rate was the highest (by 123 times from 32s to 3938s when the number of mutations increased from 50 to 500). Thus SCITE's running time was quadratic to the number of mutations. Unlike the case of increasing number of cells, BnpC was not as scalable to a large number of mutations. Its running time increased from 1199s to 8196s (by 6.8 times) when the number of mutations increased from 50 to 500. The running time growth rate for SCG and SCClone were the lowest among all except RobustClone, 2.5 times and 2.3 times, respectively. To summarize, BnpC was affected the most in terms of the running time in the

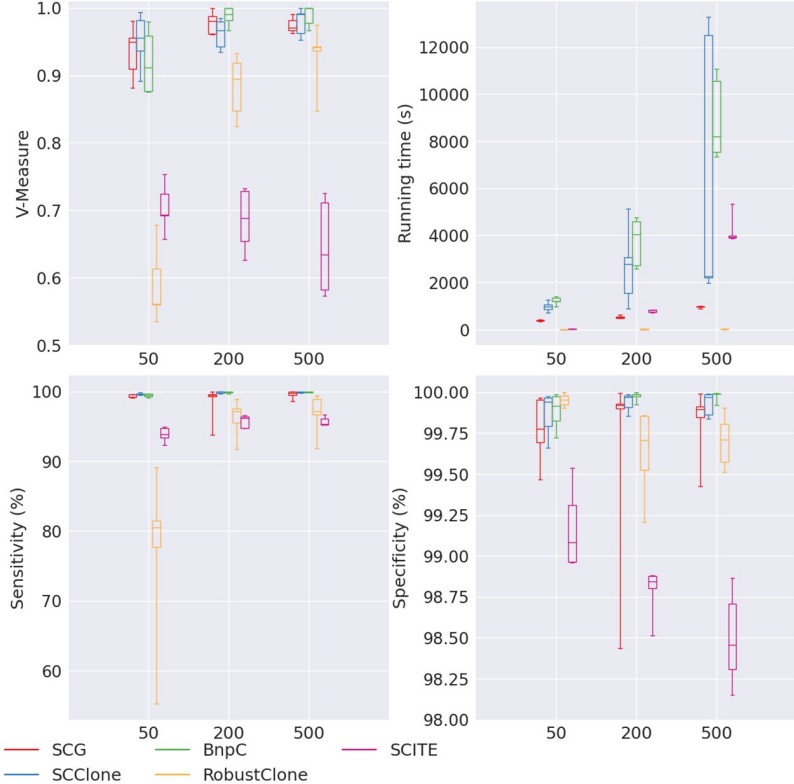

**Fig 5. Performance of SCG (red), SCClone (blue), BnpC (green), RobustClone (yellow), and SCITE (purple) for varying number of mutations, the values of which are shown on the x-axes.** The upper left, upper right, bottom left and bottom right panels are the V-measure, running time in seconds, genotyping sensitivity and genotyping specificity, respectively.

increasing number of mutations. When the number of mutations was as high as 500, it became the slowest of the five methods.

We observed decreasing V-measure, sensitivity and specificity when the number of clusters increased for all methods except SCITE (Fig 6). Of all methods, BnpC's performance was the most robust to the number of clusters. SCClone's performance was comparable to BnpC, with slightly lower specificity when the cluster number was 32. SCG was the next, whose V-measure dropped below 0.9 when the cluster number was 32. Again, we noticed that RobustClone's sensitivity dropped significantly from 100% to 73.1% when the number of clusters increased from 4 to 32 whereas its specificity stayed high (>99.7) for all numbers of clusters. Its V-measure also dropped greatly, from 1 to 0.6 when the number of clusters increased from 4 to 32. SCITE had a low V-measure (<0.8) for all numbers of clusters, and its specificity was the lowest among all methods. For this experiment of varying number of clusters, we further examined the estimation of the number of clusters from each method (Fig 7). We found that of all five methods, SCClone's estimation of the number of clusters was the most accurate. It estimated 4 and 8 clusters correctly, and 15 for 16 clusters and 28 for 32 clusters. SCG over-estimated the number of clusters for small numbers of clusters (4 or 8) and under-estimated the number of clusters for large numbers of clusters (16 or 32). In fact, its estimated number of clusters plateaued from 16 to 32 clusters. BnpC showed a trend of increasing estimated number of clusters when the true number of clusters increased and was accurate when the underlying number of clusters was small (4 and 8). However, like SCG, it also under-estimated the number of clusters

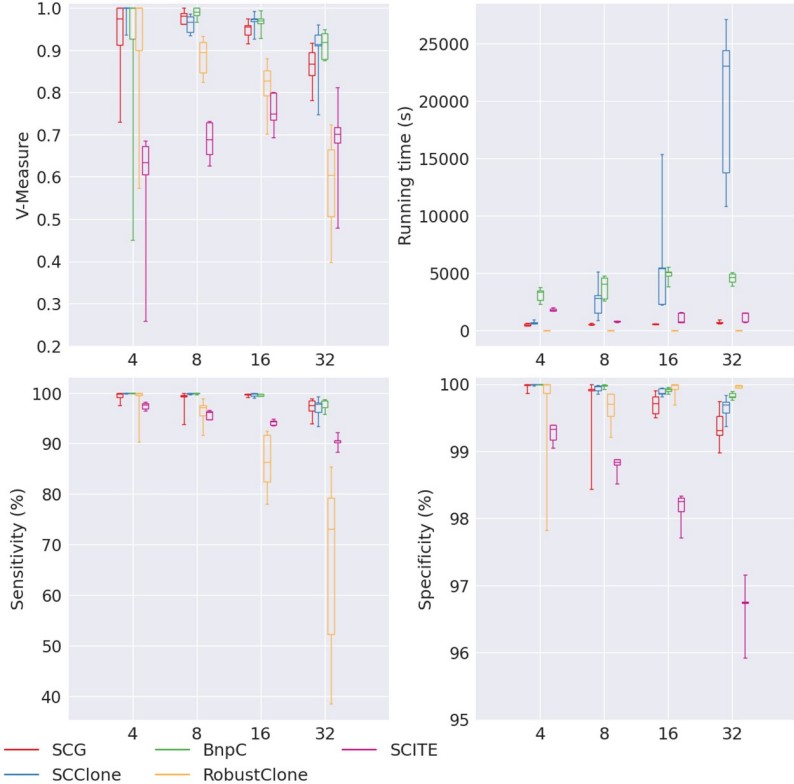

**Fig 6. Performance of SCG (red), SCClone (blue), BnpC (green), RobustClone (yellow), and SCITE (purple) for varying number of clones, the values of which are shown on the x-axes.** The upper left, upper right, bottom left and bottom right panels are the V-measure, running time in seconds, genotyping sensitivity and genotyping specificity, respectively.

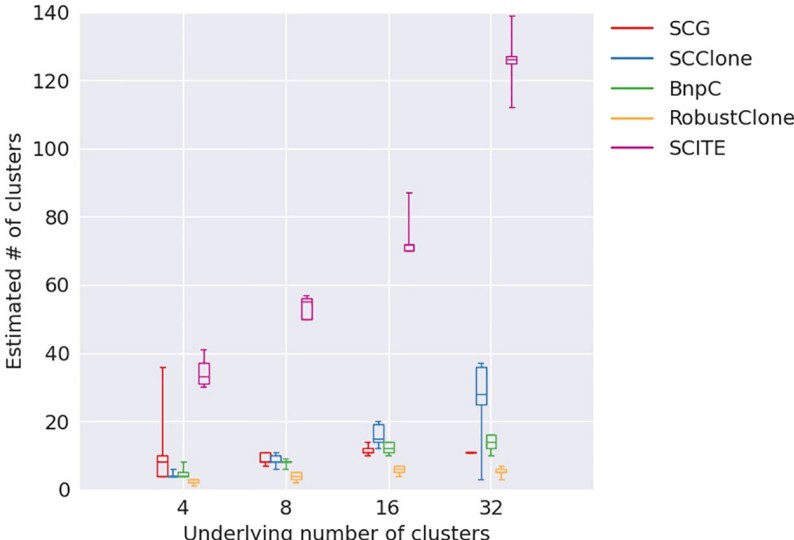

**Fig 7. Estimated number of clusters (y-axis) for varying underlying number of clusters (x-axis).**

when the underlying number of clusters was large (16 and 32). Among all five methods, RobustClone and SCITE are the worst in estimating the number of clusters. RobustClone significantly under-estimated the number of clusters for all numbers of clusters. In more detail, RobustClone estimated 3, 4, 6 and 5 as the median numbers of clusters when the true numbers of clusters were 4, 8, 16 and 32, respectively. SCITE, on the other hand, significantly over-estimated the number of clusters for all numbers of clusters. Its median estimated numbers of cluster were 33, 55, 72 and 126 when the true numbers of clusters were 4, 8, 16 and 32, respectively.

In terms of running time, RobustClone remained the most efficient, i.e., <10s for all numbers of clusters. We noticed that SCClone's running time increased significantly from 654s to 23,036s when the cluster number increases from 4 to 32. This can be explained by that SCClone increasingly searches the optimal cluster number until it finds one and thus has to run the EM algorithm for every possible cluster number. This helped SCClone to render a higher accuracy in estimating the number of clusters. SCG, BnpC and SCITE's running time were relatively stable with respect to the varying number of clusters.

In the experiment of the varying doublet rate, the input to the five methods contained doublets to different percentages. However, since we used the singlet model for SCG, and none of the rest of the four methods explicitly report doublets, we only measured V-measure, sensitivity and specificity on non-doublet cells and excluded the doublets from these measurements. Thus this experiment examines the clustering accuracy of the non-doublet cells in the face of varying doublet rate in the data.

Of all five methods, SCG, SCClone and BnpC's V-measure were the most robust to the increasing doublet rate (all >0.95), among which BnpC had the best combination of V-measure, sensitivity and specificity when the doublet rate was as high as 0.1 (Fig 8). RobustClone's V-measure remained stable near ∼0.9. Its sensitivity and specificity remained high for all doublet rates, near 96% and 99.75%, respectively. SCITE's V-measure remained the lowest (∼0.7) of all five methods for all doublet rates, with much lower specificity than other methods, ranging between 99.85% and 99%. While there was no observable trend of decreasing V-measure, sensitivity and specificity for all five methods with the increasing doublet rate probably due to

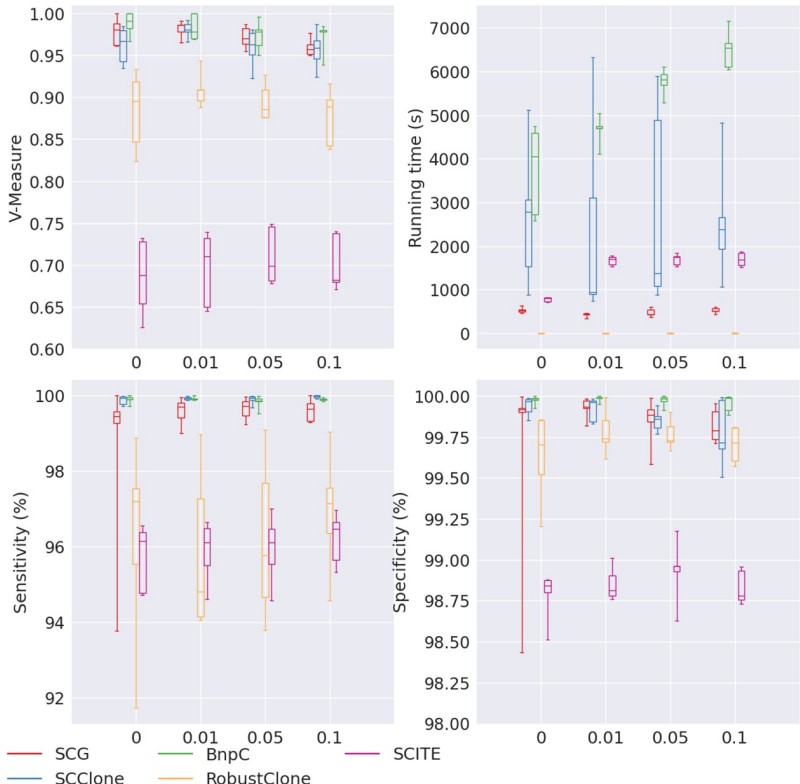

**Fig 8. Performance of SCG (red), SCClone (blue), BnpC (green), RobustClone (yellow), and SCITE (purple) for varying doublet rate, the values of which are shown on the x-axes.** The upper left, upper right, bottom left and bottom right panels are the V-measure, running time in seconds, genotyping sensitivity and genotyping specificity, respectively.

our measuring the non-doublet cells alone, it was obvious that BnpC had a trend of increasing running time (from 4046s to 6535s) when the doublet rate increased from 0 to 0.1. No other method showed any obvious trend of running time with the increasing doublet rate. Robust-Clone remained the fastest method among the five examined, whose running time was <= 10s for all doublet rates.

Lastly, we applied the five methods to the data set with varying Beta splitting variable. The Beta splitting variable was used to control the variance of the cluster sizes so that the smaller the Beta splitting variable, the bigger the variance. Specifically, in this data set, corresponding to the Beta splitting variable at 0.05, 0.2, and 0.5, the average standard deviation of the cluster sizes were 102, 71.1 and 60.32, respectively. We found that all five methods' performance dropped with the decreasing Beta splitting variable and the increasing variance of the cluster sizes (Fig 9). We noticed that when the Beta splitting variable was the smallest (0.05), SCClone, BnpC, RobustClone and SCITE's performance was not stable in the sense that some experiments in the 5 repetitions had very low V-measures. The lowest V-measure of SCClone, BnpC, RobustClone and SCITE dropped as low as 0.709, 0.458, 0.29 and 0.14, respectively, when the Beta splitting variable was as low as 0.05. SCG was relatively more stable than the other four methods when Beta splitting variable was small. Its lowest V-measure was 0.88 when Beta splitting variable was 0.05. SCG, SCClone and BnpC had high V-measure when the Beta splitting variable was the highest (0.5), all above 0.95, whereas RobustClone and SCITE's V-measure was much lower. RobustClone's V-measure was ∼ 0.9 and SCITE's was ∼ 0.7 when the Beta

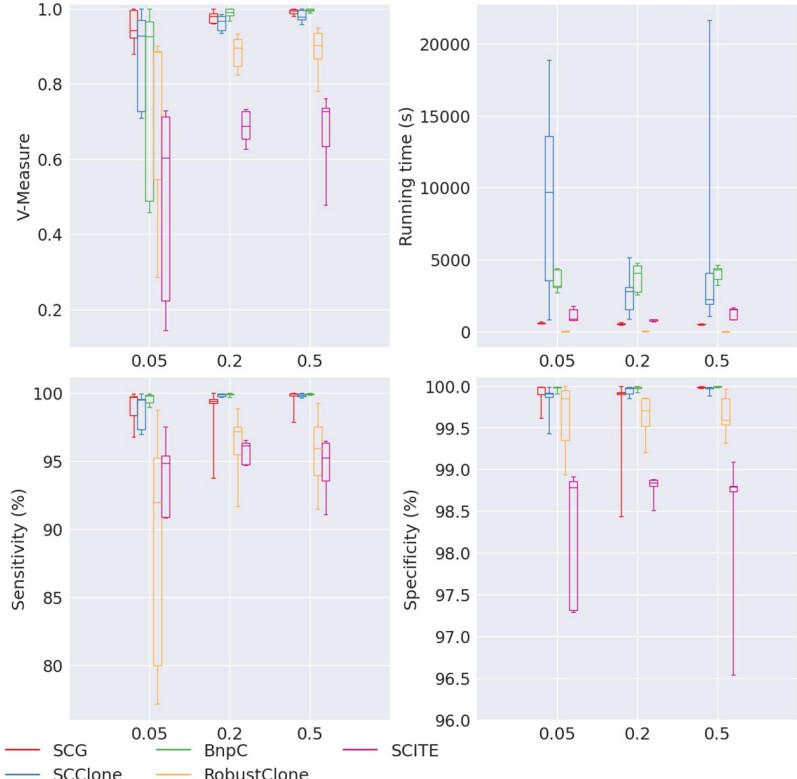

**Fig 9. Performance of SCG (red), SCClone (blue), BnpC (green), RobustClone (yellow), and SCITE (purple) for varying Beta splitting variable, the values of which are shown on the x-axes.** The upper left, upper right, bottom left and bottom right panels are the V-measure, running time in seconds, genotyping sensitivity and genotyping specificity, respectively.

splitting variable was 0.5. Similar patterns of robustness for the five methods can be observed in sensitivity and specificity as V-measure except that RobustClone's sensitivity was extremely sensitive to Beta splitting variable. Its lowest sensitivity when Beta splitting variable was 0.05 dropped down to $\sim 80\%$, to be compared with SCG's (96.78%), SCClone's (97.01%) and BnpC's (98.959%). Its sensitivity increased back to >90% when the Beta splitting variable was 0.2 and 0.5.

These observations showed that when the clone sizes did not differ from each other greatly, it was easier to cluster the cells and the clustering results were better. However, when there are both both big and small clones in a tumor, it should be expected that the clustering accuracy drops for all five methods. In terms of running time, both SCG and SCClone's median running time increased with the decrease of the Beta splitting variable. Nevertheless, SCClone's running time increased much more dramatically than that of SCG's. Specifically, its median running time increased from 2249s to 9674s, to be compared to SCG's that increased from 486s to 598s. BnpC's running time showed an opposite trend, however, whose median running time decreased from 4304s to 3179s when the Beta splitting variable decreased from 0.5 to 0.05. RobustClone's running time remained the least among all five methods (<10s) for all Beta splitting variables. SCITE's running time did not show a clear trend, and it ranged between 800s and 1600s. SCClone's increasing running time with the decrease of Beta splitting variable probably was due to the difficulty to correctly estimate the number of clusters when both small

and big clusters were present, considering that it searched for the optimal clustering solution for every increment of cluster number.

We further looked into the accuracy of the estimation of the number of clusters for the five methods for the varying Beta splitting variable (Fig 10). We performed this experiment because when Beta splitting variable is small, the variance of cluster sizes is large, and it is expected to be more challenging to correctly estimate the number of clusters. Specifically, small clusters with only a few cells are especially difficult to single out since these clusters have weak signals. Thus the signal-to-noise ratio is low in these small clusters. We found that SCG, SCClone and BnpC correctly estimated the underlying number of clusters for all Beta splitting variables, i.e., their medians equal to eight which was the correct cluster number, except that SCClone over-estimated the number of clusters when Beta splitting variable was as low as 0.05. This indicated that SCClone tends to over-segment the clusters when there are both small and large clusters in the pool. RobustClone consistently under-estimated number of clusters, whereas SCITE significantly over-estimated number of clusters. Specifically, SCITE estimated > 50 clusters for all Beta splitting variables whereas the true number of clusters was 8.

For a tabular overview of all four metrics for all varying variables, we listed Tables 1–8 here.

**Simulation results for ultra-low coverage data.** SBMClone was run on the simulated data set that had a large number of cells (4000) and mutations (5000). The data set had an ultra-low coverage that varied from 0.01 to 0.05. Since SBMClone did not report the genotype of each cell, we skipped the sensitivity and specificity evaluation. Instead, we evaluated its clustering accuracy via V-measure and the running time. The performance of SBMClone was shown in Fig 11. We found that SBMClone performed very well in terms of both V-measure (>0.95) and speed (finished within 1-2 minutes) when the coverage was 0.05. Its V-measure dropped below 0.95 when the coverage was lowered further down to 0.03, rather within a shorter running time (within about one minute). When the coverage was as low as 0.01, SBMClone's V-measure was < 0.7 (median at around 0.55), although the running time was further shortened (40-50 seconds). The decrease of both the V-measure and the running time was due to the increase of the proportion of the missing data (missing rate increased from 99.4% to 99.85% when coverage decreased from 0.05 to 0.01.).

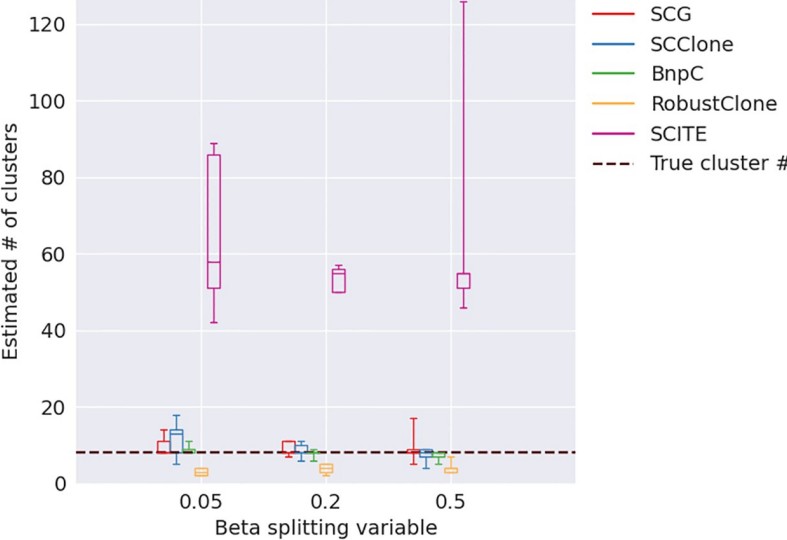

**Fig 10. Estimated number of clusters for varying Beta splitting variable.** The underlying number of clusters was 8 for all Beta splitting variables.

**Table 1. Median of sensitivity, specificity, V-measure and running time (in seconds) of the experiment with varying false positive values for SCG, SCClone, BnpC, RobustClone (RC) and SCITE.** False positive values being tested in the same order are 0.001, 0.01 and 0.05.

| FP | SCG | SCClone | BnpC | RC | SCITE |
|---|---|---|---|---|---|
| Sensitivity | 99.4 | 99.9 | 99.8 | 93.5 | 96.3 |
| | 99.4 | 99.9 | 99.9 | 97.1 | 96.1 |
| | 99.2 | 99.1 | 99.8 | 94.8 | 94.5 |
| Specificity | 100 | 100 | 100 | 99.2 | 99.2 |
| | 99.9 | 100 | 100 | 99.7 | 98.8 |
| | 100 | 99.9 | 99.2 | 99.7 | 97.4 |
| V-measure | 0.99 | 0.99 | 0.99 | 0.91 | 0.7 |
| | 0.98 | 0.97 | 0.99 | 0.89 | 0.69 |
| | 0.98 | 0.87 | 0.99 | 0.9 | 0.57 |
| Runtime | 447 | 1170 | 3461 | 5 | 1787 |
| | 522 | 2787 | 4046 | 8 | 812 |
| | 783 | 4213 | 4215 | 7 | 1576 |

**Table 2. Median of sensitivity, specificity, V-measure and running time (in seconds) of the experiment with varying false negative values for SCG, SCClone, BnpC, RobustClone (RC) and SCITE.** False negative values being tested in the same order are 0.1, 0.2, 0.3 and 0.4.

| FN | SCG | SCClone | BnpC | RC | SCITE |
|---|---|---|---|---|---|
| Sensitivity | 99.9 | 100 | 100 | 97.3 | 98.3 |
| | 99.4 | 99.9 | 99.9 | 97.1 | 96.1 |
| | 99.1 | 99.8 | 99.8 | 89.9 | 91 |
| | 99.2 | 99.8 | 99.6 | 59 | 77.5 |
| Specificity | 99.9 | 100 | 100 | 99.7 | 99 |
| | 99.9 | 100 | 100 | 99.7 | 98.8 |
| | 99.9 | 99.9 | 99.9 | 99.8 | 98.4 |
| | 99.9 | 99.8 | 99.9 | 100 | 97.8 |
| V-measure | 0.99 | 1 | 1 | 0.94 | 0.72 |
| | 0.98 | 0.97 | 0.99 | 0.89 | 0.68 |
| | 0.97 | 0.98 | 0.97 | 0.75 | 0.61 |
| | 0.97 | 0.98 | 0.96 | 0.45 | 0.56 |
| Runtime | 547 | 1281 | 4142 | 6 | 853 |
| | 522 | 2787 | 4046 | 8 | 812 |
| | 598 | 3979 | 4574 | 4 | 1662 |
| | 654 | 5246 | 4389 | 5 | 1541 |

**Table 3. Median of sensitivity, specificity, V-measure and running time (in seconds) of the experiment with varying missing rates for SCG, SCClone, BnpC, Robust-Clone (RC) and SCITE.** Missing rates being tested in the same order are 0.2 and 0.3.

| Missing rate | SCG | SCClone | BnpC | RC | SCITE |
|---|---|---|---|---|---|
| Sensitivity | 99.4 | 100 | 99.9 | 97.1 | 96.1 |
| | 99.01 | 100 | 99.81 | 94.7 | 94.7 |
| Specificity | 99.92 | 100 | 100 | 99.7 | 98.8 |
| | 99.904 | 100 | 100 | 99.7 | 98 |
| V-measure | 0.98 | 0.97 | 0.99 | 0.89 | 0.68 |
| | 0.97 | 0.98 | 0.97 | 0.84 | 0.67 |
| Runtime | 522 | 2787 | 4046 | 8 | 812 |
| | 584 | 1122 | 4177 | 7 | 1689 |

**Table 4. Median of sensitivity, specificity, V-measure and running time (in seconds) of the experiment with varying number of cells for SCG, SCClone, BnpC, RobustClone (RC) and SCITE.** Number of cells being tested in the same order are 100, 500, 1000 and 1500.

| # of cells | SCG | SCClone | BnpC | RC | SCITE |
|---|---|---|---|---|---|
| Sensitivity | 98.1 | 98.4 | 98.9 | 90.3 | 89.4 |
| | 99.4 | 99.9 | 100 | 97.1 | 96.1 |
| | 99.9 | 100 | 100 | 96.9 | 97 |
| | 100 | 100 | 100 | 94.2 | 97.8 |
| Specificity | 100 | 100 | 100 | 99.5 | 95.3 |
| | 100 | 100 | 100 | 99.7 | 98.8 |
| | 100 | 100 | 100 | 99.8 | 99.3 |
| | 100 | 100 | 100 | 99.5 | 99.6 |
| V-measure | 0.99 | 0.99 | 0.99 | 0.82 | 0.65 |
| | 0.98 | 0.97 | 0.99 | 0.89 | 0.68 |
| | 0.99 | 0.99 | 0.99 | 0.93 | 0.67 |
| | 0.99 | 1 | 1 | 0.84 | 0.7 |
| Runtime | 65 | 108 | 2793 | 3 | 392 |
| | 522 | 2787 | 4046 | 8 | 812 |
| | 2703 | 8620 | 3747 | 5 | 3380 |
| | 7178 | 12418 | 5712 | 7 | 5132 |

We intended to investigate the performance of other five tools (SCG, SCClone, BnpC, RobustClone and SCITE) on this ultra-low coverage data. To start, we simulated the ultra-low coverage data but with only 500 cells and 200 mutations. This was to test the behavior of the five tools on the data with a smaller cell number albeit much lower coverage. We expected that some methods to be extremely slow in the face of the data with such a high missing rate. We thus set a time limit of 4 hours. We found that BnpC did not finish within this time limit for the default setting. The other four methods did finish within the time limit, but they estimated either extremely low or high number of clusters. Specifically, RobustClone and SCG clustered all the cells in a single clone. SCClone clustered cells into 2 clones, whereas SCITE estimated 68 clones existed in the data, to be compared with 8 which was the true value. Moreover, RobustClone genotyped all the SNVs as 0 (reference) for all cells. Based on these observations,

**Table 5. Median of sensitivity, specificity, V-measure and running time (in seconds) of the experiment with varying number of mutations for SCG, SCClone, BnpC, RobustClone (RC) and SCITE.** Number of mutation values being tested in the same order are 50, 200 and 500.

| # of mutations | SCG | SCClone | BnpC | RC | SCITE |
|---|---|---|---|---|---|
| Sensitivity | 99.2 | 99.6 | 99.5 | 80.5 | 93.9 |
| | 99.4 | 99.9 | 99.9 | 97.1 | 96.1 |
| | 99.8 | 99.9 | 99.9 | 97.2 | 95.3 |
| Specificity | 100 | 100 | 100 | 100 | 99.1 |
| | 99.9 | 100 | 100 | 99.7 | 98.8 |
| | 100 | 100 | 100 | 99.7 | 98.5 |
| V-measure | 0.97 | 0.99 | 1 | 0.56 | 0.69 |
| | 0.98 | 0.97 | 0.99 | 0.89 | 0.68 |
| | 0.97 | 0.99 | 1 | 0.94 | 0.63 |
| Runtime | 386 | 974 | 1199 | 3 | 32 |
| | 522 | 2787 | 4046 | 8 | 812 |
| | 963 | 2258 | 8196 | 13 | 3938 |

**Table 6. Median of sensitivity, specificity, V-measure and running time (in seconds) of the experiment with varying number of clusters for SCG, SCClone, BnpC, RobustClone (RC) and SCITE.** Number of cluster values being tested in the same order are 4, 8, 16 and 32.

| # of clusters | SCG | SCClone | BnpC | RC | SCITE |
|---|---|---|---|---|---|
| Sensitivity | 99.7 | 100 | 100 | 100 | 97.6 |
| | 99.4 | 99.9 | 99.9 | 97.1 | 96.1 |
| | 99.7 | 99.8 | 99.5 | 86.3 | 94.3 |
| | 97.6 | 97.8 | 98.5 | 73.1 | 90.6 |
| Specificity | 100 | 100 | 100 | 100 | 99.3 |
| | 99.9 | 100 | 100 | 99.7 | 98.8 |
| | 99.7 | 99.9 | 99.9 | 100 | 98.3 |
| | 99.3 | 99.7 | 99.8 | 100 | 96.7 |
| V-measure | 0.98 | 1 | 1 | 1 | 0.63 |
| | 0.98 | 0.97 | 0.99 | 0.89 | 0.68 |
| | 0.96 | 0.97 | 0.97 | 0.83 | 0.75 |
| | 0.87 | 0.91 | 0.92 | 0.6 | 0.7 |
| Runtime | 482 | 654 | 3321 | 4 | 1764 |
| | 522 | 2787 | 4046 | 8 | 812 |
| | 546 | 5422 | 5043 | 6 | 745 |
| | 662 | 23036 | 4636 | 4 | 1519 |

we decided to run SCClone and SCITE on the larger ultra-low coverage data with 4000 cells and 5000 mutations to further investigate their scalability. Both SCITE and SCClone failed to finish running on this data within 48 hours. We thus do not compare SBMClone with any of these five methods on the large ultra-low coverage data.

## Real data analysis

We applied four methods, SCG, SCClone, BnpC and RobustClone, to a human colorectal cancer sample CRC2 [53]. We did not apply SBMClone to this data set due to the extremely small

**Table 7. Median of sensitivity, specificity, V-measure and running time (in seconds) of the experiment with varying doublet values for SCG, SCClone, BnpC, RobustClone (RC) and SCITE.** Doublet values being tested in the same order are 0, 0.01, 0.05 and 0.1.

| Doublets | SCG | SCClone | BnpC | RC | SCITE |
|---|---|---|---|---|---|
| Sensitivity | 99.4 | 99.9 | 99.9 | 97.1 | 96.1 |
| | 99.7 | 99.9 | 99.9 | 94.8 | 96.1 |
| | 99.7 | 99.9 | 99.8 | 95.8 | 96.1 |
| | 99.6 | 100 | 99.9 | 97.1 | 96.5 |
| Specificity | 99.9 | 100 | 100 | 99.7 | 98.8 |
| | 99.9 | 100 | 100 | 98.8 | 98.8 |
| | 99.9 | 99.9 | 100 | 99 | 98.9 |
| | 99.8 | 99.7 | 100 | 98.8 | 98.8 |
| V-measure | 0.98 | 0.97 | 0.99 | 0.89 | 0.68 |
| | 0.99 | 0.98 | 0.98 | 0.91 | 0.71 |
| | 0.97 | 0.96 | 0.98 | 0.89 | 0.7 |
| | 0.96 | 0.96 | 0.98 | 0.89 | 0.68 |
| Runtime | 522 | 2787 | 4046 | 8 | 812 |
| | 438 | 952 | 4707 | 10 | 1703 |
| | 443 | 1378 | 5805 | 7 | 1748 |
| | 512 | 2389 | 6535 | 6 | 1690 |

**Table 8. Median of sensitivity, specificity, V-measure and running time (in seconds) of the experiment with varying Beta splitting values for SCG, SCClone, BnpC, RobustClone (RC) and SCITE.** Beta splitting values being tested in the same order are 0.05, 0.2 and 0.5.

| Betasplit | SCG | SCClone | BnpC | RC | SCITE |
|---|---|---|---|---|---|
| Sensitivity | 99.7 | 99.5 | 99.8 | 92 | 94.9 |
| | 99.4 | 99.9 | 99.9 | 97.1 | 96.1 |
| | 100 | 99.8 | 99.9 | 95.9 | 95.2 |
| Specificity | 100 | 99.9 | 100 | 99.9 | 98.8 |
| | 99.9 | 100 | 100 | 99.7 | 98.8 |
| | 100 | 100 | 100 | 99.6 | 98.8 |
| V-measure | 0.94 | 0.93 | 0.93 | 0.88 | 0.6 |
| | 0.98 | 0.97 | 0.99 | 0.89 | 0.68 |
| | 1 | 0.98 | 1 | 0.9 | 0.73 |
| Runtime | 598 | 9674 | 3179 | 5 | 891 |
| | 522 | 2787 | 4046 | 8 | 812 |
| | 486 | 2249 | 4304 | 5 | 1531 |

number of cells which was not suitable for the application of SBMClone. We did not apply SCITE because SCITE's result on CRC2 has been reported in [53], whose clustering result is used in the following text as a reference.

For the cells that were suitable for SNV detection, thirty-four primary cells (denoted as PA cells) and thirty-three metastatic cells (denoted as MA cells) had been sequenced in [53] and used in this study. The distinction between primary cells and metastatic cells was based on the biological resection region of the tumor cells. We did not include the rest of the 115 normal cells that had been annotated by [53] since the focus was the tumor cells that had the mutations. According to [53], thirty-six mutations were genotyped and shown in the cell locus matrix, out of which 137 entries were missing. [53] used SCITE [27] to infer the mutation tree and the underlying genotype of the cells. SCITE inferred the existence of two metastatic clones, categorizing 15 of the 33 metastatic cells to the first metastasis (M1) and 13 to the second metastasis (M2). For the rest of the 5 MA cells, SCITE categorized them as primary cells.

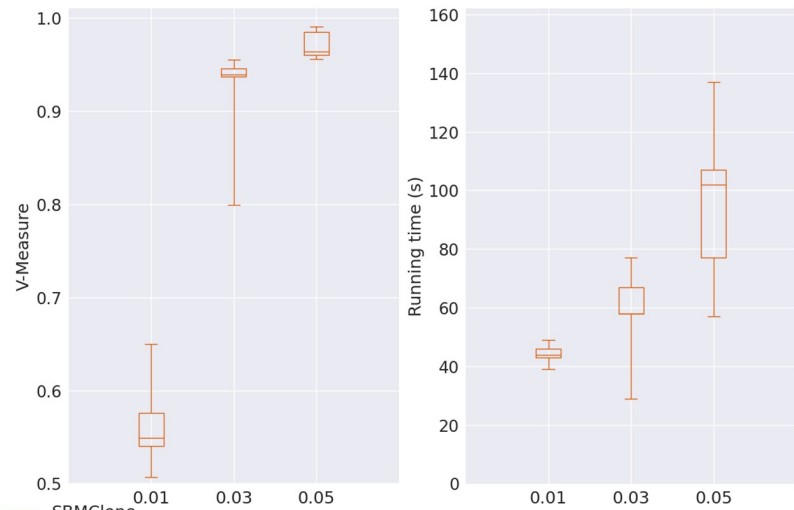

**Fig 11. V-measure and running time (in seconds) for SBMClone on coverage 0.01, 0.03 and 0.05.**

According to both SCITE and the original resection position from the patient, it was clear that in total there were three underlying clusters in this data set. SCITE also estimated the false positive and negative rates for this sample to be 0.0174 and 0.1256, respectively.

We then examined the clustering performance of SCG, SCClone, BnpC, and RobustClone on CRC2 treating SCITE's clustering result as the reference except the questionable five MA cells that had been clustered with PA cells by SCITE. Notice that we used the clustering result from SCITE as the reference not because it was the best among all clustering methods, but because SCITE's result was published together with the original paper of CRC2 and thus went through the examination from a biological point of view, as well as that SCITE's result was by large consistent with the prior knowledge of the resection positions of the cells.

We kept the parameter settings for the four methods the same as the setting for simulated data sets except increasing the "-pp" value for BnpC to 0.75 0.75 as suggested by the authors due to the decreased number of mutation sites.

We depicted the clustering results of SCG, SCClone, BnpC and RobustClone on CRC2 in Fig 12. RobustClone estimated five clusters, mixing PA, M1 and M2 cells together in almost every cluster. BnpC and SCClone had a better estimation of cluster number, both of which predicted there were four clusters in the data, whereas SCG overestimated the number of clusters to be seven. Interestingly, the five MA cells (MA39, MA41, MA42, MA44 and MA9) that were

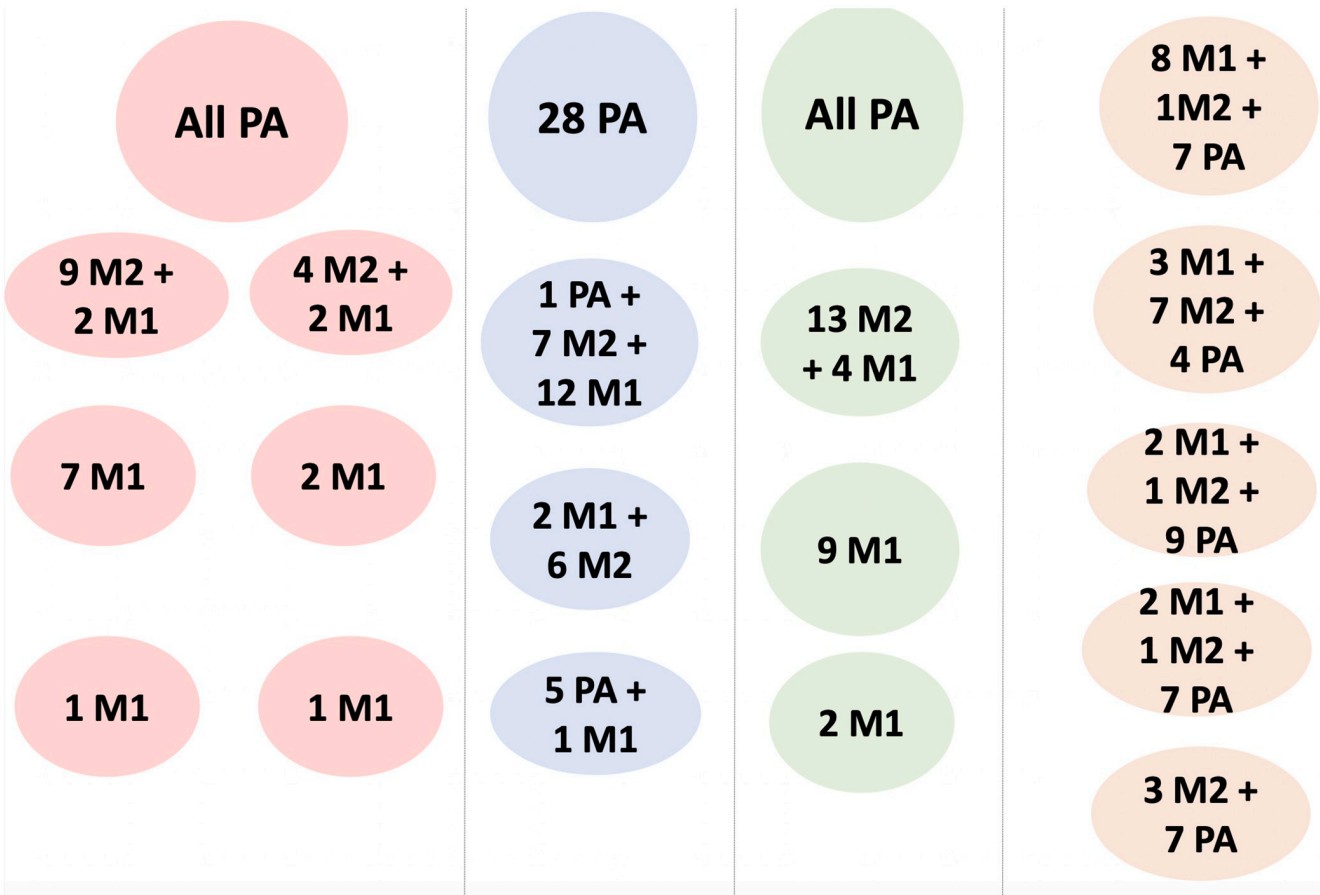

**Fig 12. Illustration of the clustering results from SCG (red background), SCClone (blue background), BnpC (green background), and RobustClone (orange background).**

categorized as primary cells by SCITE were clustered with other metastatic cells instead of primary cells by SCG, BnpC and SCClone. This indicated that clustering-based methods, due to its simplicity and smaller search dimension, may lead to a higher clustering accuracy than the methods considering the tumor lineage and clonality simultaneously. Among SCG, BnpC and SCClone, we observed that BnpC's clustering result was the most consistent with the tumor cells' original resection location as well as SCITE's inference. To be specific, we observed that BnpC and SCG clustered all the PA cells together in one cluster whereas SCClone broke the PA cells into two clusters, one of which was also mixed with a M1 cell. BnpC clustered all the M2 cells together, but mixed four M1 cells (MA27, MA48, MA88 and MA94) with this M2 cluster. SCG separated M2 cells into two clusters and also mixed the four M1 cells mentioned above, two for each, with these two M2 clusters.

BnpC clustered the remaining M1 cells into two clusters, whereas SCG broke these M1 cells into four clusters, two of which were composed of only one cell. This is consistent with the observation we found from the simulation study, where SCG over-segmented clusters when the underlying cluster number was small. Interestingly, both SCG and BnpC contained the same cluster that was composed of only two M1 cells, MA29 and MA90, showing the challenge of clustering these two cells together with other MA cells. Different from SCG and BnpC which clearly separated M1 and M2 cells, SCClone mixed up M1 and M2 cells together into two cocktail clusters. Based on these observations, we conclude that SCG tended to break large clusters into smaller ones and did not cluster all the cells together according to their original tumor location. BnpC captured the clonality of the cells according to their original tumor location. Even though SCClone's estimation of the cluster number was one of the most accurate, it's clustering result was not as accurate as BnpC. These conclusions are consistent with what we observed in the simulation study.

## Discussion

We examined five cell clustering methods for scDNAseq data that are intended for SNV detection, SCG, SCClone, BnpC, RobustClone and SCITE in terms of their clustering accuracy, specificity and sensitivity, running time and the accuracy of estimating the number of clusters based on a benchmark study and a real data sample. We also examined SBMClone's clustering accuracy and its running time on the simulated large ultra-low coverage data. In the benchmark study of the data set designed for SNV detection, we varied eight variables to comprehensively examine the performance of the five methods. In the ultra-low coverage data, we varied the coverage to examine SBMClone's performance on varying ultra-low coverages on a large data set with high numbers of cells and mutations. We summarize our observations as follows.

First of all, we found that SCITE's V-measure was consistently the lowest of all methods, ranging between 0.5 and 0.75 for most of the simulations. We observed that SCITE also tended to over-estimate the number of clusters, which was probably the reason why SCITE had low V-measure. Plus, SCITE's specificity was almost the lowest for all variables. This means that SCITE may over-correct false negative entries. Thus we suspect that instead of correcting the erroneous genotype, SCITE tended to branch out the nodes resulting in more clusters to fit the erroneous data. Another explanation is that since SCITE infers an underlying tree structure and the placement of the mutations on the tree, all the descendant nodes automatically inherit the mutations assigned to an ancestral node. This means if a mutation is assigned wrongly to an ancestor node, false positive errors will occur. The opposite situation, i.e., a mutation is assigned wrongly to a descendant node, may lead to false negative errors. Thus the underlying tree structure is a two-edged sword. It forces the ancestor-descendant or parallel relationship

among mutations, which when inferred correctly helps the clustering of the cells. However, when a mutation is placed at the wrong node, it worsens the clustering of the cells.

Second, RobustClone's running time was consistently the least among all methods probably due to that 1) it does not model the latent variables such as FP and FN rates; 2) it separately recovers the genotype matrix and clusters the cells instead of doing both jointly. The Matlab implementation of RPCA and the R-package of Louvain-Jaccard might also contribute to a much smaller running time. However, RobustClone was very sensitive to false negative rate. Its V-measure drops tremendously when false negative rate increased. In addition, we noticed that RobustClone's V-measure, sensitivity and specificity were almost always lower than those of SCG, BnpC and SCClone. This is probably due to that RobustClone does not specifically model FP and FN rate, and thus when FN rate was high, it cannot accurately correct FN entries, leading to dropping sensitivity.

Third, although SCClone's performance was comparable to SCG and BnpC for a majority of variables, SCClone was sensitive to false positive rate, whose V-measure decreased tremendously with the increase of false positive rate. This is probably because SCClone has a fixed initial FP rate. When the true FP rate was greater than the fixed initial FP rate (0.01), SCClone failed to correct those false positive entries. However, it is notable that SCClone's estimation of cluster number was the most accurate especially when the cluster number was as high as 32, although its running time dramatically increased when the cluster number increased. This is probably because SCClone performs an EM algorithm for each possible cluster number while it searches for the optimal cluster number.

We found that generally SCG and BnpC were most robust to different variables such as false positive rate, false negative rate, missing rate, number of cells, number of mutations, doublet rate and Beta splitting variable. Both SCG and BnpC's V-measure are high for these experiments. Except the experiment of the varying number of clones in which BnpC's V-measure was slightly higher than that of SCG's for clone # = 32, their V-measure were mostly very close to each other. We reason that this is because both SCG and BnpC are fully Bayesian methods that model the hidden parameters such as false positive rate and false negative rate. In addition, both methods non-parametrically model the unknown number of clones, and thus are robust to the data set with high number of clusters. In terms of running time, we found that BnpC was advantageous in their scalability to the number of cells. When the number of cells reached 1,500, BnpC's running time was smaller than that of SCG's. For all other experiments, SCG was considerably faster than BnpC probably due to the fact that SCG used a mean-field variational inference method to search the solution. We reason that BnpC's scalability to thousands of cells was probably due to their Gibbs sampling for cell assignment to a cluster, as well as their split-merge move that allows multiple cells to change their cluster assignment all at once.

It is worth to note that when the variance of the size of the clusters became larger, all five methods' performance was affected in the sense that the worst case's V-measure among the five repetitions greatly decreased. This shows that the higher the variance of the cluster sizes, the more difficult it is to cluster the cells.

Lastly, SBMClone was the only tool that generated reasonable results for the large ultra-low coverage data whose coverage ranged between 0.01 and 0.05. This is mainly due to that SBMClone uses a stochastic block model that aggregates both cell signal (cells inside the same cluster are supposed to have the same mutation profiles) and mutation signal (two mutations that occur at the same time on a phylogenetic tree are supposed to be found in the same set of cells). Thus although the missing rate was extremely high in the ultra-low coverage data, SBMClone still output results with high V-measure especially when coverage $>= 0.3$.

## Conclusion

We designed simulation experiments for both the scDNAseq data that were intended for SNV detection and the large ultra-low coverage data that were not originally intended for SNV detection. For the former experiments, we varied eight parameters to benchmark five state-of-the-art scDNAseq cell clustering methods, which are SCG, SCClone, BnpC, RobustClone and SCITE. Although SCITE also outputs the clonal tree, in this study, we focus on its cell clustering performance. For the large ultra-low coverage data, we varied the coverage and applied SBMClone to the simulated data. In addition, we applied SCG, SCClone, BnpC and RobustClone to a real data set CRC2.

We conclude that SCG is the most accurate and fastest method except when the cell number is high (>1,000) or when the estimated clone number is high (>16). BnpC's accuracy is next to SCG and it is scalable to thousands of cells. Thus we recommend BnpC when users have a data set that has thousands of cells or users have a prior knowledge that the cluster number is extremely high in the data set. SCClone is sensitive to false positive rate and is not recommended for data set that has a high false positive rate (>= 0.01). However, SCClone's estimation of the cluster number is the most accurate among all methods. Thus using SCClone for different data sets that have high and low numbers of clones may benefit the users when the number of clone itself is used for the downstream analysis, e.g., for the ITH analysis or for prognosis. Lastly, despite the fastest speed, RobustClone's performance is always lower than SCG, BnpC and SCClone. In addition, it is very sensitive to the false negative rate. Thus we do not recommended RobustClone to the community. Neither do we recommend SCITE for cell clustering due to its consistently low V-measure and over-estimating the cluster number, unless the users are interested in the mutation tree in addition to the clustering of the cells. In terms of ultra-low coverage data with thousands of cells and mutations, we highly recommend SBMClone as it is the only method that can obtain reasonable clustering result from this type of data.

## Methods

### Description of simulation

We separately examined the performance of the methods on the data that was intended for SNV detection (i.e., data from whole genome amplification methods such as MDA) and the ultra-low coverage data, the latter of which was mainly intended for CNA detection although there is a trace amount of SNV signal in the data. Such separation is necessary due to that SCG, SCClone, BnpC, RobustClone and SCITE were designed for the former type of data, whereas SBMClone was designed for the latter type of data. The major differences between these two types of data lie in 1) missing rate; 2) number of cells that might be sequenced and used. The former type of data's missing rate is as low as 0.2 or 0.3, whereas the ultra-low coverage data's missing rate is as high as >0.99. From the perspective of the number of cells sequenced, the data intended for SNV detection usually does not contain more than 500 cells in a study mainly due to the high sequencing cost, whereas thousands of the cells can be sequenced in the ultra-low coverage data thanks to the much lower sequencing cost per cell. In the following text, we describe our simulation for both types of data.

**Simulate data intended for SNV detection.** To fully examine the performance of SCG, SCClone, BnpC, RobustClone and SCITE, we designed a set of simulated data. For each simulated data set, we first simulated a clonal tree on which each leaf node represented a subclone of cells. Thus the number of leaves was the true number of clusters in cell clustering. The branches on the clonal tree were simulated such that each branch followed an exponential

distribution with λ equaled to 1. The simulator distributed a given total number of mutations, $M$, to all the branches according to their lengths. The tree structure was simulated in the following way. Starting from the root node that was free from any mutations, we split a leaf node in the tree. When there was only a root node, it had to be selected to split. Once a node was split, it became the parent node of the two newly formed leaf nodes and itself was no longer a leaf node. Each node was given an interval whose start and end were within 0 and 1, and whose length was ≤ 1. The root node's interval is [0, 1]. The union of the two leaf nodes' intervals was the same as that of the parent node's interval, i.e., the child nodes split the parent node's interval. The ratio of the interval length of the two child nodes was decided by a sampling from the Beta distribution whose $\alpha$ value was fixed as 0.5 and $\beta$ value was a variable that we varied in a set of simulated data. This variable was referred to as "Beta splitting variable". When the Beta splitting variable is 0.5, the chance the left and right daughter nodes share evenly the parent node's interval is the highest. When the Beta splitting variable decreases ($< 0.5$), the left daughter node starts to have a higher chance to share a bigger proportion of the parent node's interval. More explanation of the importance of the interval length can be found in the following text. One of the purposes to have an interval corresponding to each node is that the chance of a leaf node being selected to split is in proportion to one minus its interval length. Notice that in our previous approach SimSCSnTree [51], the chance of a leaf node being selected to split is in proportion to the interval length, instead of one minus the interval length. This is a subtle but key difference because our "reverse splitting" strategy allows leaf nodes with smaller intervals to have a higher chance to split. The second purpose of having the interval corresponding to each node is that after generating the tree structure, we assigned a given number of cells, $N$, to all the leaf nodes in proportion to their interval lengths. Thus the interval length also decides the cluster size. In such sense, the "reverse splitting" strategy retains the size of big clusters, but small clusters continue to split, resulting in a larger contrast of the cluster sizes. This procedure helped us to effectively use the Beta splitting variable to control the contrast of the sizes of the clusters and further allowed us to examine different methods' accuracy when cluster sizes differed to different degrees. The whole splitting process stopped once the number of leaf nodes reached the desired number. Once the splitting process was completed and the tree structure had been formed, for each edge, we sampled its edge length from an exponential distribution whose λ is 1. Afterwards we assigned a given number of cells to the leaf nodes in proportion to each leaf node's interval length, and a given number of SNVs to all edges on the tree in proportion to each edge's length.

Once the cells were assigned to leaf nodes and the SNVs to the edges, for each leaf node $x$, we walked the path from the root node to $x$ and assigned the SNVs on the branches along the path to the cells assigned to $x$. We then formed the genotype matrix $G$ which was a cell locus matrix with the cells on the rows and SNV loci on the columns. We assigned the values of matrix $G$ based on the following: $G_{i,j} = 1$ if SNV $j$ was present in cell $i$ and $G_{i,j} = 0$ if SNV $j$ was absent in cell $i$. We used a binary genotype matrix instead of a ternary one so that all of the five methods can be fairly compared. Notice that $G$ matrix was the underlying true genotype matrix without any error or missing data. For a cell $i$, $G_i$ was in fact the same as the true underlying consensus genotype for the leaf node (or cluster, as each leaf node represents a cluster) that cell $i$ was assigned to. We then imputed the missing data, false positives and false negatives on the $G$ matrix to get the resulting noisy data matrix ($D$ matrix). In the real scenario, $D$ matrix is what can be observed from the sequencing data, whereas $G$ matrix is the desired matrix to be inferred from the computational tools. In imputing the missing data, since we used the number "3" to represent the missing data, we randomly turned an entry, no matter whether its value was a 0 or 1, to be 3, according to a pre-set missing data probability. In imputing the false positive entries and false negative entries, of all the entries that were 0 and 1, we flipped

them to be 1 and 0 according to a pre-set false positive rate and false negative rate, respectively. Since all five methods being evaluated on the data set simulated in the above-mentioned way (SCG, SCClone, BnpC, RobustClone and SCITE) take the noisy cell locus matrix as the input, the resulting simulated matrix ($D$ matrix) was the input to these tools. Notice that an alternative way to simulate the input to these methods is to simulate the read counts supporting the variant and reference alleles, respectively, for each SNV, and then convert the read count to the $D$ matrix according to the number of variant-supporting reads and total number of reads. This way of simulation is more realistic because it takes into account the coverage of the data and sequencing errors. Although this way mimics the real sequencing process better, we decided not to implement the simulator in this way because the former way that simulates $D$ matrix from $G$ matrix directly can give us a much better control of the false positive rate, false negative rate and missing rate. Thus we can produce a $D$ matrix that has exactly the desired false positive rate, false negative rate and missing rate. On the other hand, turning the read count to a $D$ matrix is nontrivial because it depends on the SNV calling algorithms. Some SNV calling algorithms, such as Monovar [54], utilize the read count from a group of cells to detect SNVs, and thus are sensitive in calling the SNVs even if some cells have a small number of variant-supporting reads, whereas other SNV calling algorithms, such as GATK HaplotypeCaller [55], are less sensitive at the SNVs that have low variant-supporting reads. However, the discussion of the upstream SNV calling algorithm is out of the scope of this study. Since all the five methods being evaluated take the $D$ matrix instead of the read count as the input, to gain a better control of the error rates and not to dive deep into the differences of the SNV calling algorithms, we did not use the read count to determine the $D$ matrix but directly generated the $D$ matrix from the $G$ matrix.

**Simulating ultra-low coverage data.** For the study of the performance of SBMClone, we simulated the large ultra-low coverage data. The first few steps, such as simulating the tree structure, imputing the mutations on the tree edges, assigning cells to the tree leaf nodes, and obtaining the genotype G matrix, were exactly the same procedures as we did to simulate the data intended for SNV detection. However, since in ultra-low sequencing data, whether a SNV can be detected heavily depends on the number of variant-supporting reads covering the SNV site, the number of total and variant-supporting reads shall be simulated before the D matrix can be obtained. The total and variant-supporting read numbers, on the other hand, heavily depends on the coverage of the data. We therefore simulated the total and variant-supporting read numbers, $t_{ij}$ and $v_{ij}$ for cell $i$ at SNV site $j$, based on the true genotype $G_{ij}$ and a given coverage $c$, the latter of whose value was varied to discover SBMClone's performance in the face of data with different coverages. Notice that since we were focusing on the ultra-low coverage data in this experiment, all coverages being investigated were $<< 1$. To simulate the total read number, we randomly sampled a number $x \in [0, 1]$. We assigned the total read number $t_{ij}$ based on the following: If $c^2 \le x < c$, $t_{ij} = 1$; if $x < c^2$, $t_{ij} = 2$; otherwise $t_{ij} = 0$. We did not go further down for the cases when there were three or more than three reads covering a SNV site because such a situation was extremely impossible for the ultra-low coverage data. Simulating variant-supporting read number, on the other hand, depended on the value of $G_{ij}$. When $G_{ij} = 0$, the chance that a read covering the SNV site being a variant-supporting read is very low and depends on the sequencing error rate. Here we set the sequencing error rate to be 0.01. Thus we stochastically determined each read in the total read count to be a variant-supporting read with 1% of the chance. When $G_{ij} = 1$, we assume all SNV sites were heterozygous and we considered the cases with or without ADO. First of all, we fixed the ADO rate to be 0.2. We stochastically determined that ADO did happen with 20% of the chance. If we determined ADO did happen, we then randomly determined whether the ADO occurred to the reference allele or the variant allele. If the reference allele was determined to be dropped out, all the reads

covering this SNV site were variant-supporting reads. If the variant allele was determined to be dropped out, all the reads covering this SNV site were reference-supporting reads, leading to zero variant-supporting reads. If ADO was determined not to happen, the chance that a read supported the variant was 0.5. Thus for each read covering this SNV site, we randomly determined this read to be a variant- or reference-supporting read.

After obtaining the total and variant-supporting read numbers, we obtained the observed genotype matrix $D$ in the following way: if $v_{ij} \geq 1$, $D_{ij} = 1$; otherwise $D_{ij} = 3$. Thus only when there was at least one variant-supporting read can a SNV site be observed. In all other situations, we counted this SNV site as missing. The reason is, even if the total read number is $> 0$ and the variant-supporting read number is zero, we are not sure whether the zero variant-supporting read number is due to the lack of coverage or there is no variant allele to be sequenced from.

## The design of simulation experiments

Since our simulation procedures were different for the data intended for SNV detection and the ultra-low coverage data that was not originally intended for SNV detection, we designed their simulation experiments accordingly and they are described below in two different subsections.

**Simulation experiments for the data intended for SNV detection.** For the simulated data intended for SNV detection, we benchmarked the five methods (SCG, SCClone, BnpC, RobustClone and SCITE) given eight varying parameters, which were false positive rates, false negative rates, missing rates, number of cells, number of mutations, cluster number (or leaf node number), doublet rate and the Beta splitting variable that controlled the variance of cluster sizes. Beta splitting variable has never been investigated in the previous study and thus is unique in this study.

For each variable, we set up a default value. For each experiment, we varied only one variable while setting all other variables to be the default value, as seen in Table 9. For false positive rate, we selected the range according to [29, 30, 53]. For false negative rate, we selected the range according to [22, 56–59]. For doublet rates, we selected the range according to [19, 29]. Notice that since BnpC and SCClone do not explicitly model doublets, we set the default doublet rate to be zero. To further examine how the existence of doublet cells may affect the accuracy of clustering, we varied doublet rates between 1% and 10%. In terms of varying cell numbers, we set up a big range (100 to 1500) to test the algorithms since BnpC claimed to be scalable to a large number of cells. Similar to the number of cells, we set up a big range (50 to 500) of mutations to test the accuracy as well as the running time of all five methods.

**Table 9. Simulated datasets for the data intended for SNV detection.** Each line has a varying variable (first column) with the values in the second column. The default value is denoted by "(d)" on its right.

| | |
|---|---|
| False positive rate | 0.001, 0.01 (d), 0.05 |
| False negative rate | 0.1, 0.2 (d), 0.3, 0.4 |
| Missing rate | 0.2 (d), 0.3 |
| # of cells | 100, 500 (d), 1000, 1500 |
| # of mutations | 50, 200 (d), 500 |
| Cluster number | 4, 8 (d), 16, 32 |
| Doublet rate | 0 (d), 0.01, 0.05, 0.1 |
| Beta splitting variable | 0.05, 0.2 (d), 0.5 |

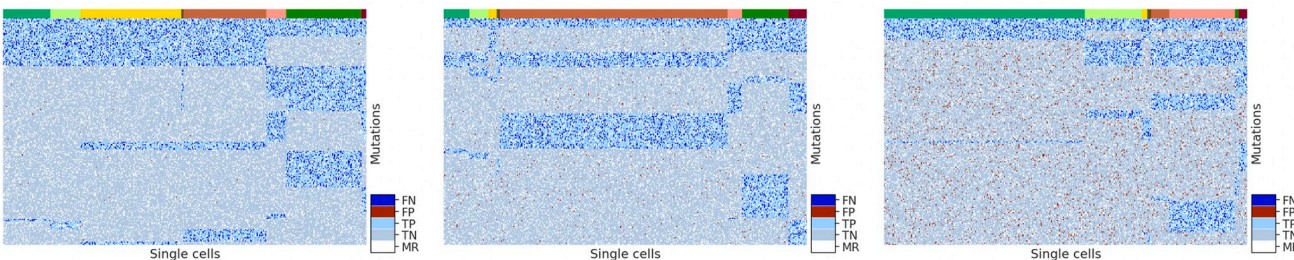

**Fig 13. Heatmaps of a randomly sampled simulated *D* matrix whose false positive rate is 0.001 (leftmost), 0.01 (middle) and 0.05 (rightmost) whereas all other parameters are default.** For each heatmap, the rows represent mutations and the columns represent cells. The top color bar represents different subclones of cells. Cells corresponding to the same subclone are clustered together under the same color in the color bar. Inside the heatmap, there are five colors of the dots showing false negative (dark blue), false positive (dark red), true positive (light blue), true negative (grey), and missing (white) entries.

For each such combination of variables, we repeatedly created the tree structure, assigned the mutations and cells, imputed the errors on the *G* matrix and created the *D* matrix for five times to overcome the random extremity.

To show that our tree-based simulation method and the parameter setting are representative of the *D* matrices generated from the actual scDNA-seq data, we randomly sampled the simulated *D* matrix and showed the corresponding heatmaps for varying false positive rates and false negative rates, respectively, in Figs 13 and 14. Here we specifically emphasized these two error rates because they are not immediately observed by a user.

**Simulation experiments for ultra-low coverage data.** To simulate ultra-low coverage data, the most important variable is the coverage. We therefore varied the coverage from 0.01 to 0.05. We increased the number of cells to 4000 and the number of mutations to 5000 in this set of data thanks to the decreasing sequencing cost in the ultra-low coverage data. We fixed the number of true clones present in this ultra-low coverage data as 8. For each coverage, we repeated the whole process for five times to overcome the random extremity.

## Comparison metrics

To compare the clustering results, we used four different metrics, and they are V-measure, running time, sensitivity and specificity. V-measure measures the clustering results given the ground truth, whereas sensitivity and specificity measure the consensus genotype inference. In particular, sensitivity measures the percentage of correctly reported 1's by the methods over the total number of 1's in the ground truth *G* matrix, and specificity measures the percentage of correctly reported 0's by the methods over the total number of 0's in the ground truth *G* matrix.

We also measured the genotyping accuracy which is the percentage of correctly reported 1's and 0's by the methods over the total number of entries in the ground truth *G* matrix, but decided not to include it in this manuscript because of its redundancy with the sensitivity and specificity. In addition to these four metrics, we also evaluated the accuracy of the estimated number of clusters for two experiments, the varying number of clusters and the varying Beta splitting variable. Whether the five methods can correctly estimate the number of clusters on these two experiments is interesting because when the underlying cluster number or the variance of cluster sizes is large, the estimation of the cluster number is challenging.

## Usage of SCG, SCClone, BnpC, RobustClone, SCITE and SBMClone

We tested all the methods by running them alternatively on either of these two types of processors: Intel(R) Xeon(R) CPU E5-2680 V4 @ 2.40GHz and Intel(R) Xeon(R) Gold

6248 CPU @ 2.50GHz. The parameter setting to run each of the methods is described as follows.

- For SCG, we set the maximum number of clusters *num_clusters* to be *N*/4 where *N* is the number of cells. The number of maximum iteration −*max_num_iters* was set to be $10^6$, the *kappa_prior* was set to 1 and the *gamma_prior* was set as "[9.99, 0.01, 1.0e-15][2.5, 7.5, 1.0e-15][1.0e-15, 1.0e-15, 1]". We used "[1, 1, 1.0e-15]" for the *state_prior* and set the number of restart to be 20 with a random seed value −*seed* selected from 0 to 10000. After running SCG for 20 restarts, we chose the best seed value depending on the lower bound value, as suggested in [44]. We then used this best seed value and the above-mentioned parameters for a final run for SCG, whose results were taken as the final result. We used the most recent version published in 2021 of the singlet model in GitHub for our experiments.

- For SCClone, we used default parameters in which false positive rate $\alpha$ was initialized to be 0.01 and false negative rate $\beta$ was selected from the grid search. We used the recent version published in 2022 available in GitHub for our experiments.

- We ran BnpC by the default setting in all our simulated data sets. In the real data, we set Beta function shape parameters *pp* value to be "0.75 0.75" as suggested by the authors since our

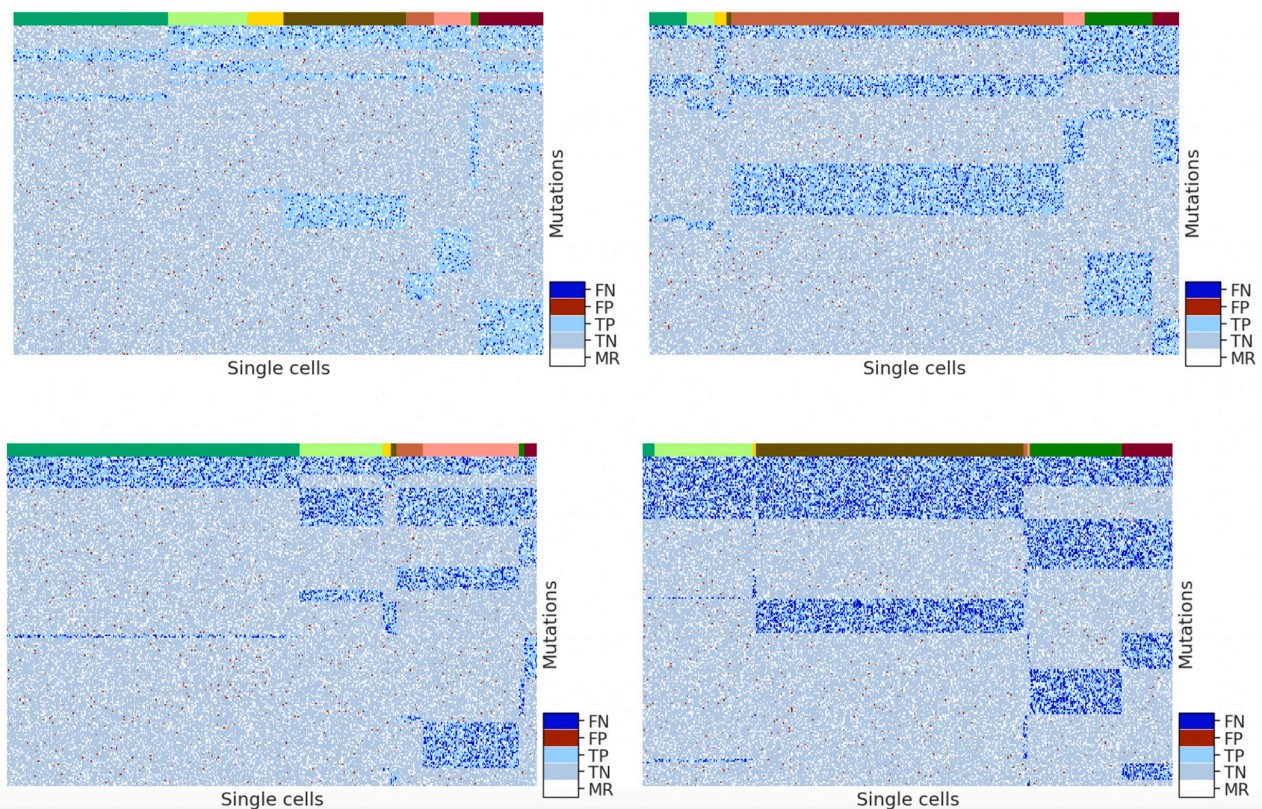

**Fig 14. Heatmaps of a randomly sampled simulated *D* matrix whose false negative rate is 0.1 (upper left), 0.2 (upper right), 0.3 (bottom left) and 0.4 (bottom right) whereas all other parameters are default.** For each heatmap, the rows represent mutations and the columns represent cells. The top color bar represents different subclones of cells. Cells corresponding to the same subclone are clustered together under the same color in the color bar. Inside the heatmap, there are five colors of the dots showing false negative (dark blue), false positive (dark red), true positive (light blue), true negative (grey), and missing (white) entries.

real data set had less than 40 mutations. We used the version available in GitHub before their October 2022 update for our experiments.

- For RobustClone, we used their default setting in all our simulated data sets and real data. We used the version published in 2020 available in GitHub for our experiments. We wrapped their MATLAB and R scripts so that the input and output files can be passed as arguments as these were not available in the downloaded version.

- For SCITE, suggested by the authors, we set the desired number of repetitions of the MCMC $-r$ = 1, desired chain length of each MCMC repetition $-l$ = $10n^2 \log n$ in which $n$ represents the number of mutations. Further, we set $-max\_treelist\_size$ = 1 to obtain one co-optimal tree as an output and used the option $-a$ to obtain the list of cells attached to each leaf node. Finally, we set $-fd$ = 0.01 and $-ad$ = 0.2, and $-n$ and $-m$ to be the known value of mutations and cells respectively. We used the most recent version published in 2018 available in GitHub for our experiments.

- For SBMClone, we used their default parameters in all our simulated data sets. We used the most recent version published in 2021 available in GitHub for our experiments.

Since SCG does not directly output the cluster assignment for each cell but provides the posterior probability of each cell belonging to each cluster, we obtained the cluster assignment as follows. For each cell, we selected the cluster that the cell had the highest posterior probability with. If there were two clusters that had the tied highest probability for a cell, we randomly selected one of the clusters. Likewise, since SCG does not directly output the consensus genotype for each cluster but provides the posterior probability of a mutation being a certain genotype in a cluster, we binarized such a probability into the consensus genotype in the following way. For each inferred cluster and each mutation, we obtained the genotype by choosing the one with the highest posterior probability. If there were two genotypes that had the tied highest probability, we used the genotype that was supported by the most cells assigned to this cluster.

We intended to run D-SCG on our simulated data sets since D-SCG is doublet-aware. However, it takes extremely long time (longer than one day) to finish the jobs. We reasoned this is because D-SCG's computational complexity is $O(NK^2M)$, in which $K$ is the number of clusters and $M$ is the number of mutations. Thus D-SCG is quadratic with the number of clusters. We then switched back to SCG and measured the clustering accuracy only for the non-doublet cells. Thus all five methods are not doublet-aware and the measurements do not include doublet cells throughout the entire benchmark study. However, the accuracy of clustering non-doublet cells in the presence of different levels of doublet cells is still of interest and we included this study in one of our simulation experiments, in which the input to the five methods involved doublets, whereas the evaluation metrics (V-measure, sensitivity and specificity) did not involve doublets.

Since SCITE does not directly output the consensus genotype, we obtained it as follows. For each cluster, we obtained the consensus genotype by the majority vote of all cells belonging to this cluster based on their $D$ matrix. For those missing entries in the $D$ matrix, we randomly assigned 0 or 1. Notice that all cells belonging to the same cluster will have the same corrected genotype profile, which is the consensus genotype of the cluster they belong to.

## Acknowledgments

We would like to thank Dr. Andrew Roth who helped us tune SCG's configuration parameters. We would like to thank Dr. Nico Borgsmüller for helping us understand the output of BnpC. We would like to thank Dr. Ziwei Chen for helping us extract the clonal genotype matrix from

RobustClone's output. We would like to thank Dr. Jack Kuipers for helping us with the proper parameter setting of SCITE and obtaining the cell clustering results.

## Author Contributions

**Conceptualization:** Xian Mallory.

**Data curation:** Rituparna Khan.

**Funding acquisition:** Xian Mallory.

**Investigation:** Rituparna Khan, Xian Mallory.

**Methodology:** Rituparna Khan, Xian Mallory.

**Project administration:** Xian Mallory.

**Resources:** Xian Mallory.

**Software:** Rituparna Khan, Xian Mallory.

**Supervision:** Xian Mallory.

**Validation:** Rituparna Khan, Xian Mallory.

**Visualization:** Rituparna Khan.

**Writing – original draft:** Rituparna Khan, Xian Mallory.

**Writing – review & editing:** Xian Mallory.

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
