## [Decision Letter · Decision Letter 0]

5 Dec 2022

Dear Dr. Mallory,

Thank you very much for submitting your manuscript "Assessing the Performance of Methods for Cell Clustering from Single-cell DNA Sequencing Data" for consideration at PLOS Computational Biology.

As with all papers reviewed by the journal, your manuscript was reviewed by members of the editorial board and by several independent reviewers. In light of the reviews (below this email), we would like to invite the resubmission of a significantly-revised version that takes into account the reviewers' comments.

Please note that this was a borderline decision, and the revision needs to address all concerns raised by the reviewers. In particular, for a benchmarking paper it is important to broadly include all types of methods rather than stating that some are known to perform worse - the point of the benchmarking is to quantitatively evaluate that. 

We cannot make any decision about publication until we have seen the revised manuscript and your response to the reviewers' comments. Your revised manuscript is also likely to be sent to reviewers for further evaluation.

Sincerely,

Bjoern Peters

Section Editor

PLOS Computational Biology

Bjoern Peters

Section Editor

PLOS Computational Biology

Reviewer's Responses to Questions

**Comments to the Authors:**

Reviewer #1: Khan and Mallory report a benchmarking study for 3 methods for clustering single—cell data. The study is well designed and should be helpful for users of such clustering methods in selecting a method for their data.

In the introduction the authors claim that cell clustering methods are more accurate than tree reconstruction methods but they don’t provide any references. Was this proven before?

When describing the simulator, the authors mention that each node was assigned an interval. What do the intervals represent? What is the beta splitting variable? Were the SNV assigned randomly?

The authors need to describe the computing resources that were used for the reader to understand the described runtimes.

Was Cross—validation performed when calculating performance metrics? This should be done and also clarified.

A tabular overview of all performance metrics and all simulations would be helpful.

The differences of the underlying algorithms for each method should be discussed. Do those differences explain the difference in performance?

Reviewer #2: This paper assesses three methods for cell clustering and genotyping of SNVs measured in single-cell DNA sequencing data. To do so, the authors adapted their previous simulator to better control clone sizes. I have a couple of concerns that reduced my enthusiasm for this paper.

0. Selection of methods

It is unclear why tree based methods are excluded. bnpc for example includes a comparison to SiCloneFit. Also, there is a recent method called SBMClone [1] that was not included in the comparison. Finally, the authors write:

"Here we skip discussing RobustClone because it does not have a friendly user interface and is not as popular."

I suggest rephrasing this ("not as popular").

1. Writing needs to be improved.

There are several grammatical errors throughout.

2. Sensitivity and specificity are undefined

It is unclear how these two metrics are defined. Is it in terms of pairs of cells? Please elaborate. Also, please elaborate why these two metrics render genotype accuracy redundant?

3. Doublets

Papers states:

"Thus all three methods are not doublet-aware and the measurements do not include doublet cells throughout the entire benchmark study. However, the accuracy of clustering non- doublet cells in the presence of different levels of doublet cells is still of interest and we included this study in one of our simulation experiment."

It is unclear whether doublets were excluded from the input, or only excluded when computing V-measure? This needs to be clarified.

4. Effect of coverage not assessed

My biggest concern is that the starting point of the simulations is a binary matrix indicating presence/absence of mutations in individual cells. This is unrealistic, as in practice one observes variant and reference read counts. Moreover, with more recent technologies the coverage is low (0.01-0.05X). I recommend the authors to also assess ultra-low coverage data.

[1] Myers, Matthew A, Simone Zaccaria, and Benjamin J Raphael. “Identifying Tumor Clones in Sparse Single-Cell Mutation Data.” Bioinformatics 36, no. Supplement_1 (July 1, 2020): i186–93. https://doi.org/10.1093/bioinformatics/btaa449.

**Have the authors made all data and (if applicable) computational code underlying the findings in their manuscript fully available?**

Reviewer #1: Yes

Reviewer #2: Yes

PLOS authors have the option to publish the peer review history of their article (what does this mean?). If published, this will include your full peer review and any attached files.

Reviewer #1: No

Reviewer #2: No
---

## [Decision Letter · Decision Letter 1]

11 Jul 2023

Dear Dr. Mallory,

Thank you very much for submitting your manuscript "Assessing the Performance of Methods for Cell Clustering from Single-cell DNA Sequencing Data" for consideration at PLOS Computational Biology.

As with all papers reviewed by the journal, your manuscript was reviewed by members of the editorial board and by several independent reviewers. In light of the reviews (below this email), we would like to invite the resubmission of a significantly-revised version that takes into account the reviewers' comments. **Please also find comments from an external Editor review below this email.**

We cannot make any decision about publication until we have seen the revised manuscript and your response to the reviewers' comments. Your revised manuscript is also likely to be sent to reviewers for further evaluation.

Sincerely,

Jason Papin

Editor-in-Chief

PLOS Computational Biology

**External Editor Review Comments: **

I've carefully reviewed the revised manuscript and reviews. I believe that I see the main issue and how it can be resolved. The authors respond to all of reviewer 1's suggestions. The authors respond fully including additional analyses as well as relevant changes to the text to all of reviewer 2's suggestions/critiques except "4. Effect of coverage not assessed" and "2. Sensitivity and specificity are underlined". For these two suggestions/critiques, the authors make clarifications in the text but rebut major aspects of the suggestions/critiques. The most significant of these is "4. Effect of coverage not assessed" where reviewer 2 starts this critique with "My biggest concern...". I will address this first, followed by 2. which is minor. I'll end with my brief thoughts on the manuscript as it is.

Regarding reviewer 2's point 4, reviewer 2 has a point that what a user would observe/have in hand is variant and reference read counts. The reviewer suggests simulating that as opposed to what the authors simulate and is a next step in the analysis of this kind of data which is generating a D matrix containing inferred genotypes from the read count data. The authors argue that the D matrix is what's input to each of the tools they're assessing, so they stand by simulating the D matrix and not read counts. I see both points. Here's my view of this standoff. As long as the D matrices that the authors are simulating are representative of the D matrices that would be generated by actual scDNA-seq data, then I believe the arguments the authors make are valid and their results are useful to the scDNA-seq community. They would have to redo the whole analysis from a new starting point if they had to simulate the variant/reference read counts. However, the authors don't go into great detail regarding the parameters that they used to simulate their D matrices and evaluate all of these tools. These parameters are shown in Table 9 (and used throughout the text) and the values they take are justified in one sentence in the methods (on p. 29 of the current text): "We selected the ranges for each variable mainly according to the discussion in [19]". These parameters include false positive and false negative rate which are not immediately observed by a user which supports, in part, reviewer 2's issue with this approach. The authors go into great detail about their tree-based synthetic data generation method which also appears to follow a previous publication. As long as this tree-based method and the parameter settings are representative of actual D matrices generated from actual scDNA-seq data then the authors benchmarking and conclusions should be on reasonably solid ground and useful to the scDNA-seq community. For example, they have two clear winning methods/tools, BnpC and SCG, across all parameters that they explored. So the conclusions are relatively simple and don't depend on read coverage or any of the simulated parameters (except cell number which is directly observable and affects runtime of these top two tools). If not, then reviewer 2's concerns are valid. One way to address this in a more conclusive manner, than just relying on reference [19], is to demonstrate that the simulated D matrices are representative of/effectively indistinguishable from D matrices derived from actual scDNA-seq data. This should at least be true for parameters within the range of those shown in Table 9. This could be done by randomly sampling these matrices and generating heatmaps or taking another approach. This would resolve the issue in my view.

Regarding reviewer 2's point 2, the authors argue that they effectively assess accuracy which depends on sensitivity and specificity so they show sensitivity and specificity separately without also explicitly calculating accuracy. The authors are correct. Formally, Accuracy = Sensitivity * Prevalence + Specificity * (1 - Prevalence) where Prevalence = Positives/(Positives + Negatives). By showing sensitivity and specificity separately, they show the weakest of the two which allows a bound of the Accuracy without knowing the Prevalence. The two top methods, BnpC and SCG, tend to have high values of both across parameters assessed, so this is not an issue.

If the authors demonstrate more convincingly, that the D matrices that they generate are representative of actual D matrices derived from actual scDNA-seq data, then I think they were responsive to reviews and that this paper would be useful to the scDNA-seq community.

Reviewer's Responses to Questions

**Comments to the Authors:**

Reviewer #1: All my concerns have been addressed

**Have the authors made all data and (if applicable) computational code underlying the findings in their manuscript fully available?**

Reviewer #1: None

PLOS authors have the option to publish the peer review history of their article (what does this mean?). If published, this will include your full peer review and any attached files.

Reviewer #1: No
---

## [Decision Letter · Decision Letter 2]

20 Sep 2023

Dear Dr. Mallory,

We are pleased to inform you that your manuscript 'Assessing the Performance of Methods for Cell Clustering from Single-cell DNA Sequencing Data' has been provisionally accepted for publication in PLOS Computational Biology.

Best regards,

Edwin Wang

Section Editor

PLOS Computational Biology

Bjoern Peters

%CORR_ED_EDITOR_ROLE%

PLOS Computational Biology

Reviewer's Responses to Questions

**Comments to the Authors: **

Reviewer #3: The authors have addressed my comments/suggestions.

**Have the authors made all data and (if applicable) computational code underlying the findings in their manuscript fully available?**

Reviewer #3: Yes

PLOS authors have the option to publish the peer review history of their article (what does this mean?). If published, this will include your full peer review and any attached files.

Reviewer #3: **Yes: **Stefan Bekiranov

---

## [Editor Report · Acceptance letter]

9 Oct 2023

PCOMPBIOL-D-22-01229R2 

Assessing the Performance of Methods for Cell Clustering from Single-cell DNA Sequencing Data

Dear Dr Mallory,

I am pleased to inform you that your manuscript has been formally accepted for publication in PLOS Computational Biology. Your manuscript is now with our production department and you will be notified of the publication date in due course.

With kind regards,

Zsuzsanna Gémesi
